# USING LATENT SPACE REGRESSION TO ANALYZE AND LEVERAGE COMPOSITIONALITY IN GANS

**Lucy Chai, Jonas Wulff & Phillip Isola**
MIT CSAIL, Cambridge, MA 02139, USA
{lrchai,wulff,phillipi}@mit.edu

## ABSTRACT

In recent years, Generative Adversarial Networks have become ubiquitous in both research and public perception, but how GANs convert an unstructured latent code to a high quality output is still an open question. In this work, we investigate regression into the latent space as a probe to understand the compositional properties of GANs. We find that combining the regressor and a pretrained generator provides a strong image prior, allowing us to create composite images from a collage of random image parts at inference time while maintaining global consistency. To compare compositional properties across different generators, we measure the trade-offs between reconstruction of the unrealistic input and image quality of the regenerated samples. We find that the regression approach enables more localized editing of individual image parts compared to direct editing in the latent space, and we conduct experiments to quantify this independence effect. Our method is agnostic to the semantics of edits, and does not require labels or predefined concepts during training. Beyond image composition, our method extends to a number of related applications, such as image inpainting or example-based image editing, which we demonstrate on several GANs and datasets, and because it uses only a single forward pass, it can operate in real-time. Code is available on our project page: https://chail.github.io/latent-composition/.

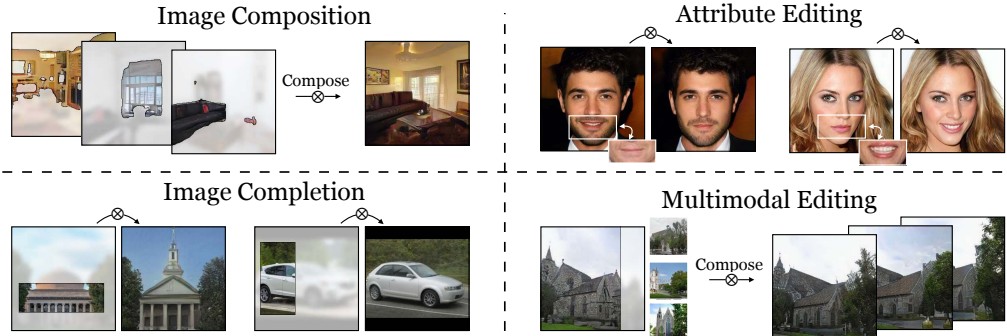

Figure 1: Simple latent regression on a fixed, pretrained generator can perform a number of image manipulation tasks based on single examples without requiring labelled concepts during training. We use this to probe the ability of GANs to compose scenes from image parts, suggesting that a compositional representation of objects and their properties exists already at the latent level.[1]

## 1 INTRODUCTION

Natural scenes are comprised of disparate parts and objects that humans can easily segment and interchange (Biederman, 1987). Recently, unconditional generative adversarial networks (Karras et al., 2017; 2019b;a; Radford et al., 2015) have become capable of mimicking the complexity of natural images by learning a mapping from a latent space noise distribution to the image manifold. But how does this seemingly unstructured latent space produce a strikingly realistic and structured

---

[1]Dome image from: https://www.technologyreview.com/2019/10/24/132370/mit-dome/

scene? Here, we use a latent regressor to probe the latent space of a pretrained GAN, allowing us to uncover and manipulate the concepts that GANs learn about the world in an unsupervised manner.

For example, given a church image, is it possible to swap one foreground tree for another one? Given only parts of the building, can the missing portion be realistically filled? To achieve these modifications, the generator must be compositional, i.e., understanding discrete and separate representations of objects. *We show that the pretrained generator – without any additional interventions – already represents these compositional properties in its latent code.* Furthermore, these properties can be manipulated using a regression network that predicts the latent code of a given image. The pixels of this image then provide us with an intuitive interface to control and modify the latent code. Given the modified latent code, the network then applies image priors learned from the dataset, ensuring that the output is always a coherent scene regardless of inconsistencies in the input (Fig. 1).

Our approach is simple – given a fixed pretrained generator, we train a regressor network to predict the latent code from an input image, while adding a masking modification to learn to handle missing pixels. To investigate the GAN's ability to produce a globally coherent version of a scene, we hand the regressor a rough, incoherent template of the scene we desire, and use the two networks to convert it into a realistic image. Even though our regressor is never trained on these unrealistic templates, it projects the given image into a reasonable part of the latent space, which the generator maps onto the image manifold. This approach requires no labels or clustering of attributes; all we need is a single example of approximately how we want the generated image to look. It only requires a forward pass of the regressor and generator, so there is low latency in obtaining the output image, unlike iterative optimization approaches that can require upwards of a minute to reconstruct an image.

We use the regressor to investigate the compositional capabilities of pretrained GANs across different datasets. Using input images composed of different image parts ("collages"), we leverage the generator to recombine this unrealistic content into a coherent image. This requires solving three tasks simultaneously – blending, alignment, and inpainting. We then investigate the GAN's ability to independently vary localized portions of a given image. In summary, our contributions are:

- We propose a latent regression model that learns to perform image reconstruction even in the case of incomplete images and missing pixels and show that the combination of regressor and generator forms a strong image prior.

- Using the learned regressor, we show that the representation of the generator is already compositional in the latent code, without having to resort to intermediate layer activations.

- There is no use of labelled attributes nor test-time optimization, so we can edit images based on a single example of the desired modification and reconstruct in real-time.

- We use the regressor to probe what parts of a scene can vary independently, and investigate the difference between image mixing using the encoder and interpolation in latent space.

- The same regressor setup can be used for a variety of other image editing applications, such as multimodal editing, scene completion, or dataset rebalancing.

## 2 RELATED WORK

**Image Inversion.** Given a target image, the GAN inversion problem aims to recover a latent code which best generates the target. Image inversion comes with a number of challenges, including 1) a complex optimization landscape and 2) the generator's inability to reconstruct out-of-domain images. To relax the domain limitations of the generator, one possibility is to invert to a more flexible intermediate latent space (Abdal et al., 2019), but this may allow the generator to become overly flexible and requires regularizers to ensure that the recovered latent code does not deviate too far from the latent manifold (Pividori et al., 2019; Zhu et al., 2020; Wulff & Torralba, 2020). An alternative to increasing the flexibility of the generator is to learn an ensemble of latent codes that approximate a target image when combined (Gu et al., 2019a). Due to challenging optimization, the quality of inversion depends on good initialization. A number of approaches use a hybrid of a latent regression network to provide an initial guess of the latent code with subsequent optimization of the latent code (Bau et al., 2019; Guan et al., 2020) or the generator weights (Zhu et al., 2016; Bau et al., 2020; Pan et al., 2020), while Huh et al. (2020) investigates gradient-free approaches for optimization. Besides inverting whole images, a different use case of image inversion through a generator is to complete

partial scenes. When using optimization, this is achieved by only measuring the reconstruction loss on the known pixels (Bora et al., 2017; Gu et al., 2019a; Abdal et al., 2020), whereas in feed-forward methods, the missing region must be provided explicitly to the model. Rather than inverting to the latent code of a pretrained generator, one can train the generator and encoder jointly, based on modifications to the Variational Autoencoder (Kingma & Welling, 2013). Donahue et al. (2017); Donahue & Simonyan (2019); Dumoulin et al. (2016) use this setup to investigate the properties of latent representations learned during training, while Pidhorskyi et al. (2020) demonstrate a joint learning method that can achieve comparable image quality to recent GAN models. In our work, we investigate the *emergent priors of a pretrained GAN* using a masked latent regression network as an approximate image inverter. While such a regressor has lower reconstruction accuracy than optimization-based techniques, its lower latency allows us to investigate the learned priors in a computationally efficient way and makes real-time image editing incorporating such priors possible.

**Composition in Image Domains.** To join segments of disparate image sources into one cohesive output, early works use hand-designed features, such as Laplacian pyramids for seamless blending (Burt & Adelson, 1983). Hays & Efros (2007) and Isola & Liu (2013) employ nearest-neighbor approaches for scene composition and completion. More recently, a number of deep network architectures have been developed for compositional tasks. For discriminative tasks, Tabernik et al. (2016) and Kortylewski et al. (2020) train CNNs with modified compositional architectures to understand model interpretability and reason about object occlusion in classification. For image synthesis, Mokady et al. (2019) and Press et al. (2020) use an autoencoder to encode, disentangle, and swap properties between two sets of images, while Shocher et al. (2020) mixes images in deep feature space while training the generator. Rather than creating models specifically for image composition or scene completion objectives, we investigate the ability of a pre-trained GAN to mix-and-match parts of its generated images. Related to our work, Besserve et al. (2018) estimates the modular structure of GANs by learning a casual model of latent representations, whereas we investigate the GAN's compositional properties using image inversion. Due to the imprecise nature of image collages, compositing image parts also involves inpainting misaligned regions. However, in contrast to inpainting, in which regions have to be filled in otherwise globally consistent images (Pathak et al., 2016; Iizuka et al., 2017; Yu et al., 2018; Zeng et al., 2020), the composition problem involves correcting inconsistencies as well as filling in missing pixels.

**Image Editing.** A recent topic of interest is editing images using generative models. A number of works propose linear attribute vector editing directions to perform image manipulation operations (Goetschalckx et al., 2019; Jahanian et al., 2020; Shen et al., 2019; Kingma & Dhariwal, 2018; Karras et al., 2019a; Radford et al., 2015). It is also possible to identify concepts learned in the generator's intermediate layers by clustering intermediate representations, either using segmentation labels (Bau et al., 2018) or unsupervised clustering (Collins et al., 2020), and change these representations to edit the desired concepts in the output image. Suzuki et al. (2018) use a spatial feature blending approach which mixes properties of target images in the intermediate feature space of a generator. On faces, editing can be achieved using a 3D parametric model to supervise the modification (Tewari et al., 2017; 2020). In our work, we do not require clusters or concepts in intermediate layers to be defined a priori, nor do we need distinct input and output domains for approximate collages and real images, as in image translation tasks (Zhu et al., 2017; Almahairi et al., 2018). Unlike image manipulation using semantic maps (Park et al., 2019; Gu et al., 2019b), our approach respects the style of the manipulation (e.g. the specific color of the sky), which is lost in the semantic map representation. Our method shares commonalities with Richardson et al. (2020), although we focus on investigating compositional properties rather than image-to-image translation. In our approach, we only require a single example of the approximate target property we want to modify and use regression into the latent space as a fast image prior to create a coherent output. This allows us to create edits that are not contingent on labelled concepts, and we do not need to modify or train the generator.

## 3 METHOD

### 3.1 LATENT CODE RECOVERY IN GANS

GANs provide a mapping from a predetermined input distribution to a complex output distribution, e.g. from a standard normal $\mathcal{Z}$ to the image manifold $\mathcal{X}$, but they are not easily invertible. In other

words, given an image sample from the output distribution, it is not trivial to recover the sample from the input distribution that generated it. The image inversion objective aims to find the latent code $z$ of GAN $G$ that best recovers the desired target image $x$:

$$z^* = \arg\min_z(\text{dist}(G(z),\ x)), \tag{1}$$

using some metric of image distance $\text{dist}$, such as pixel-wise $L_1$ error or a metric based on deep features. This objective can be solved iteratively, using L-BFGS (Liu & Nocedal, 1989) or other optimizers. However, iterative optimization is slow – it takes a large number of iterations to converge, is prone to local minima, and must be performed for each target image $x$ independently.

An alternative way of recovering the latent code $z$ is to train a neural network to directly predict it from a given image $x$. In this case, the recovered latent code is simply the result of a feed-forward pass through a trained regressor network, $z^* = E(x)$, where $E$ can be used for any $x \in \mathcal{X}$. To train the regressor (or encoder) network $E$, we use the latent encoder loss

$$\mathcal{L} = \mathbb{E}_{z \sim N(0,1),\ x=G(z)} \left[ ||x - G(E(x))||_2^2 + \mathcal{L}_p(x, G(E(x))) + \mathcal{L}_z(z, E(x)) \right]. \tag{2}$$

We sample $z$ randomly from the latent distribution, and pass it through a pretrained generator $G$ to obtain the target image $x = G(z)$. Between the target image $x$ and the recovered image $G(E(x))$, we use a mean square error loss to guide reconstruction and a perceptual loss $\mathcal{L}_p$ (Zhang et al., 2018) to recover details. Between the original latent code $z$ and the recovered latent code $E(x)$, we use a latent recovery loss $\mathcal{L}_z$. We use mean square error or a variant of cosine similarity for latent recovery, depending on the GAN's input normalization. Additional details can be found in Supp. Sec. A.1.1.

Throughout this paper the generators are frozen, and we only optimize the weights of the encoder $E$. When using ProGAN (Karras et al., 2017), we train the encoder network to directly invert to the latent code $z$. For StyleGAN (Karras et al., 2019b), we encode to an expanded $\mathcal{W}^+$ latent space (Abdal et al., 2019). Once trained, the output of the latent regressor yields a latent code such that the reconstructed image looks perceptually similar to the target image.

## 3.2 LEARNING WITH MISSING DATA

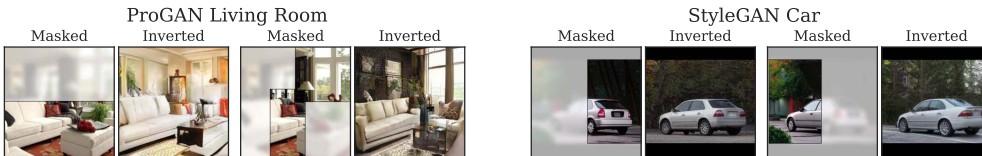

Figure 2: Image completions using the latent space regressor. Given a partial image, a masked regressor realistically reconstructs the scene in a way that is consistent with the given context. The completions ("Inverted") can vary depending on the exposed context region of the same input.

When investigating the effect of localized parts of the input images, we might want to treat some image regions explicitly as "unknown", either to create buffer zones to avoid seams between different pasted parts or to explicitly let the image prior fill in unknown regions. In optimization approaches using Eqn. 1, this can be handled by optimizing only over the known pixels. However, a regressor network cannot handle this naively – it cannot distinguish between unknown pixels and known pixels, and will try to fit the values of the unknown pixels. This can be mitigated with a small modification to the regression network, by indicating which pixels are known versus unknown as input (Fig. 3):

$$\mathcal{L} = \mathbb{E}_{z \sim N(0,1),\ x=G(z)} ||x - G(E(x_m, m))||_2^2 + \mathcal{L}_p(x, G(E(x_m, m))) + \mathcal{L}_z(z, E(x_m, m)) \tag{3}$$

Instead of taking an image $x$ as input, the encoder takes a masked image $x_m$ and a mask $m$, where $x_m = x \otimes m$, and $m$ is an additional channel of input. Intuitively, this masking operation is analogous to "dropout" (Srivastava et al., 2014) on pixels – it encourages the encoder to learn a flexible way of recovering a latent code that still allows the generator to reconstruct the image. Thus, given only partial images as input, the encoder is encouraged to map from the known pixels to a latent code that is semantically consistent with the rest of the image. This allows the generator to reproduce an image that is both likely under its prior and consistent with the observed region.

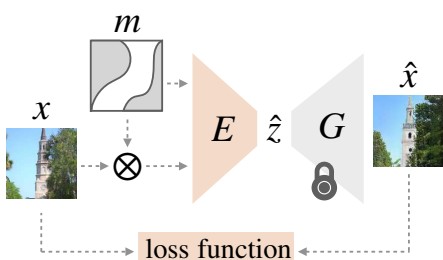

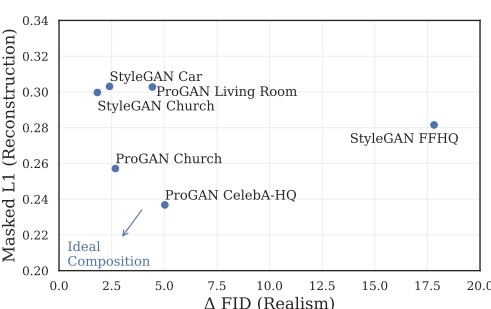

Figure 3: We train a latent space regressor $E$ to predict the latent code $\hat{z}$ that, when passed through a fixed generator, reconstructs input $x$. At training and test time, we can also modify the encoder input with additional binary mask $m$ (Eqn. 3). Inference requires only a forward pass and the input $x$ can be unrealistic, as the encoder and generator serve as a prior to map the image back to the image manifold.

Figure 4: Composition of unrealistic input collages is a balance of two factors: we want to reconstruct the input (low $L_1$), but still retain realistic images (low FID). Using automatic collages of synthesized image parts, we plot this tradeoff of masked $L_1$ error between the input collages and output composites, and FID change between the output composites and re-encoded images on different image domains.

To obtain the masked image during training we take a small patch of random uniform noise $u$, upsample this noise to the full resolution using bilinear interpolation, and mask out all pixels where the upsampled noise is smaller than a sampled threshold $t \sim \mathcal{U}(0, 1)$ to simulate arbitrarily shaped mask boundaries. However, at test time, the exact form of the mask does not matter – the mask simply indicates where the generator should try to reconstruct or inpaint, and does not distinguish between the different image parts of the input. We provide additional details in Supp. Sec. A.1.1 and A.2.3.

The regressor and generator pair enforces global coherence: when we obscure or modify parts of the input, the generator will create an output that is still overall consistent. By masking out arbitrary parts of the image (Eqn. 3), we allow the GAN to imagine a realistic completion of the missing pixels, which can vary based on the given context (Fig. 2). This suggests that the regressor inherently learns an unsupervised object representation, allowing it to complete objects from only partial hints even though the generator and regressor are never provided with structured concept labels during training.

### 3.3 IMAGE COMPOSITION USING LATENT REGRESSION

The regressor $E$ and generator $G$ form an image prior to project any input image $x_{\text{input}}$ onto the manifold of generated images $\mathcal{X}$, even if $x_{\text{input}} \notin \mathcal{X}$. We leverage this to investigate the compositional properties of the latent code. We extract parts of images (either generated by $G$ or from real images), and combine them to form a collaged image $x_{\text{clg}}$. This extraction process does not need to be precise and can have obvious seams and missing pixels. At the same time, while $x_{\text{clg}}$ is often not realistic, our encoder is aware of these missing pixels and can properly process them, as described in Sec. 3.2. We can therefore use the $E$ and $G$ to blend the seams and produce a realistic composite output. To create $x_{\text{clg}}$, we sample base images $x_i$ and masks $\text{mask}_i$, and combine them via union; once we have formed the collage $x_{\text{clg}}$, we reproject via the regressor and generator to obtain the composite $x_{rec}$:

$$x_{\text{clg}} = \bigcup_i \text{mask}_i \otimes x_i;$$
$$x_{\text{rec}} = G(E(x_{\text{clg}}, \cup_i \text{mask}_i)). \tag{4}$$

Note that each $\text{mask}_i$ used to extract individual image parts in Eqn. 4 is not available to the encoder, only the union is provided in the form of a binary mask. Also, the regressor is trained solely for the latent recovery objective (Eqn. 3) and has never seen collaged images during training. To automate the process of extracting masked images, we use a pretrained segmentation network (Xiao et al., 2018) and sample from the output classes (see Supp. Sec. A.1.2). However, the masked regressor is agnostic to how image parts are extracted; we also experiment with a saliency network (Liu et al., 2018), approximate rectangles, and user-defined masks in Supp. Sec. A.2.1 and A.2.4.

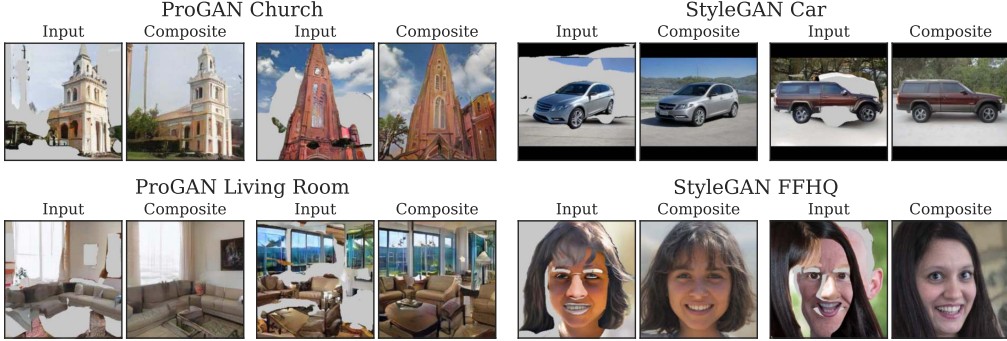

Figure 5: Trained only on a masked reconstruction objective, a regressor into the latent space of a pretrained GAN allows the generator to recombine components of its generated images, despite strong misalignments and missing regions in the input. Here, we show automatically generated collaged inputs from extracted image parts and the corresponding outputs of the generators.

## 4 EXPERIMENTS

Using pre-trained Progressive GAN (Karras et al., 2017) and StyleGAN2 (Karras et al., 2019b) generators, we conduct experiments on CelebA-HQ and FFHQ faces and LSUN cars, churches, living rooms, and horses to investigate the compositional properties that GANs learn from data.

### 4.1 IMAGE COMPOSITION FROM APPROXIMATE COLLAGES

The masked regressor and the pretrained GAN form an image prior which converts unrealistic input images into realistic scenes while maintaining properties of the input image. We use this property to investigate the ability of GANs to recombine synthesized collages; i.e., to join parts of different input images into a coherent composite output image. The goal of a truly "compositional" GAN would be to both preserve the input parts and unify them realistically in the output. As we do not have ground-truth composite images, we create them automatically using randomly extracted image parts. The regressor and generator must then simultaneously 1) blend inconsistencies between disparate image parts 2) correct misalignments between the parts and 3) inpaint missing regions, balancing global coherence of the output image with its similarity to the input collage.

Using extracted and combined image parts (Eqn. 4), we show qualitative examples of these input collages and the corresponding generated composite across a variety of domains (Fig. 5); note that the inputs are not realistic, often with imperfect detections and misalignments. However, the learned image prior from the generator and encoder fixes these inconsistencies to create realistic outputs.

To measure the tradeoff between the networks' ability to preserve the input and the realism of the composite image, we compute masked $L_1$ distance as a metric of reconstruction (lower is better)

$$\mathrm{avg}(m \otimes |x - G(E(x, m))|) \tag{5}$$

and FID score (Heusel et al., 2017) over 50k samples as a metric of image quality (lower is better). To isolate the realism of the composite image from the regressor network's native reconstruction capability (*i.e.* the ability to recreate a single image generated by G), we compare the difference in FID between the reconstructed composites ($x_{rec}$ in Eqn. 4), and re-encoded images $G(E(G(z)))$. In Fig. 4, we plot these two metrics for both ProGAN and StyleGAN across various dataset domains. Here, an ideal composition would attain zero $L_1$ error (perfect reconstruction of the input) and zero FID increase (preserves realism), but this is impossible, hence the generators demonstrate a balance of these two ideals along a Pareto front. We show full results on FID, density & coverage (Naeem et al., 2020), and $L_1$ reconstruction error and additional random samples in Supp. Sec. A.2.4.

### 4.2 COMPARING COMPOSITIONAL PROPERTIES ACROSS ARCHITECTURES

Given approximate and unrealistic input collages, the combination of regressor and generator imposes a strong image prior, thereby correcting the output so that it becomes realistic. How much does the

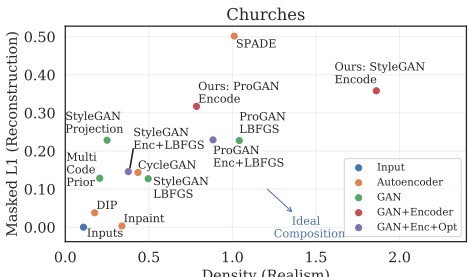 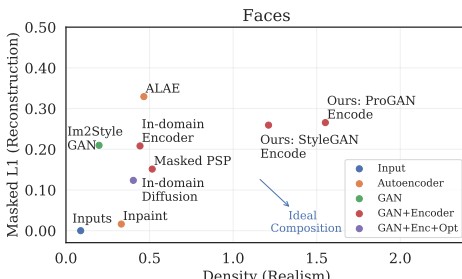

Figure 6: Comparing reconstruction of image collages (masked $L_1$) to realism of the generated outputs on random church collages (left) and face collages (right) across different image reconstruction methods, broadly characterized as autoencoders, GAN-based optimization, GANs with an encoder to perform latent regression, and a combination of GAN, regression, and optimization. An ideal composition has low $L_1$ and high density (close to real image manifold), and each method exhibits different tradeoffs in reconstruction and realism.

pretrained GAN and the regression network each contribute to this outcome? Here, we investigate a number of different image reconstruction methods spanning three major categories: autoencoder architectures without a pretrained GAN, optimization-based methods of recovering a GAN latent code without an encoder, and encoder-based methods paired with a pretrained GAN. For comparison, we use the same set of collages to compare the methods, generated from parts of random real images of the church and face domains. As some methods take several minutes to reconstruct a single image, we use 200 collages for each domain. Due to the smaller sample size, we use density here as a measure of realism (higher is better), which measures proximity to the real-image manifold (Naeem et al., 2020) and compare to $L_1$ reconstruction (Eqn. 5); a perfect composite has high density and low $L_1$. We report additional metrics in Tab. 4-5.

For the church domain, we first compare to autoencoding methods that train the generator and encoder portions jointly: DIP (Ulyanov et al., 2018), Inpainting (Yu et al., 2018), CycleGAN (Zhu et al., 2017), and SPADE (Park et al., 2019). For iterative optimization methods using only the pretrained generator, we compare direct LBFGS optimization of the latent code (Liu & Nocedal, 1989), Multi-Code Prior (Gu et al., 2019a), and StyleGAN projection (Karras et al., 2019a). Third, we use our regressor network to directly predict the latent code in a feed-forward pass (Encode), and additionally further optimize the predicted latent to decrease reconstruction error (Enc+LBFGS). We provide additional details on each method in Supplementary Sec. A.2.4. Qualitatively, the different methods have varying degrees of realism when trying to reconstruct unrealistic input collages (we show examples in Supp. Fig. 18); optimization-based methods such as Deep Image Prior, Multi-Code Prior, and StyleGAN projection tend to overfit and lead to unrealistic reconstructions with low density, whereas segmentation-based methods such as SPADE are not trained to reconstruct the input, leading to high $L_1$. Our StyleGAN encoder yields the most realistic composites with highest density, at the cost of distorting the unrealistic inputs. Fig 6-(left) illustrates this compositional tradeoff, where the encoder based methods perform slightly worse in $L_1$ reconstruction compared to optimization approaches, but maintain more realistic output and can reconstruct with lower computational time.

On the face domain, we compare the realism/reconstruction tradeoff of composite outputs of optimization-based Im2StyleGAN (Abdal et al., 2019), Inpainting (Yu et al., 2018), autoencoder ALAE (Pidhorskyi et al., 2020), and different regression networks including In-Domain Inversion (Zhu et al., 2020), Pixel2Style2Pixel (Richardson et al., 2020), and our regressor networks. We show qualitative examples in Supplementary Fig. 19 and a comparison of reconstruction $L_1$ and density in Fig. 6-(right): our ProGAN and StyleGAN masked encoders can maintain closer proximity to the real image manifold (higher density) compared to the alternative methods, with much faster inference time compared to optimization-based methods such as Im2StyleGAN. On these same inputs, ALAE exhibits interesting compositional properties and is qualitatively able to correct misalignments in face patches, but the density of generated images is lower than that of the pretrained GANs. Again, no method can achieve both reconstruction and realism perfectly due to the imprecise nature of the input, and each method demonstrates different balances between the two factors.

### 4.3 HOW DOES COMPOSITION DIFFER FROM INTERPOLATION?

Combining images in pixel space and using the encoder bottleneck to rectify the input is only one way that a generator can mix different images. Another commonly used method is to simply use a

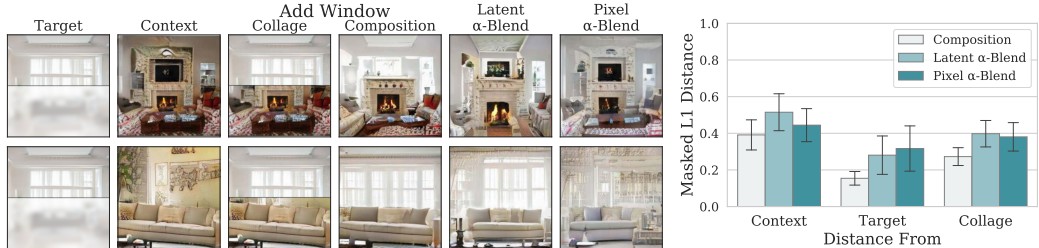

Figure 7: Comparing composition and interpolation. We aim to apply the same target modification (white windows; Target), to two context sources (Context), where the collage of the two images is shown in the third column (Collage). Compared to Latent and Pixel $\alpha$-blending, inverting the collages into the latent space via the encoder (Composition) better matches the context and target regions, while at the same time ensuring global coherence between the target and context images.

linear interpolation between two latent codes to obtain an output that has properties of both input images. Here, we investigate how these two approaches differ. When composing parts of two images, we desire that *part of a context image stays the same while the remaining portion changes to match a chosen target*: to achieve this *composition*, we select the desired pixels from our context image and the target modification, and pass the result through the encoder to obtain the blended image.

$$G(E(m_1 \otimes x_1 + m_2 \otimes x_2)) \qquad \triangleleft \quad \textit{composition} \qquad (6)$$

How does this compare to directly interpolating in latent space? We compute the *latent $\alpha$-blend* by performing a weighted average of the context and target latent codes:

$$G(\alpha * E(x_1) + (1 - \alpha) * E(x_2)) \qquad \triangleleft \quad \textit{latent $\alpha$-blend} \qquad (7)$$

and the *pixel $\alpha$-blend* by blending inputs in pixel space and using the encoder bottleneck to make the output more realistic:

$$G(E(\alpha * x_1 + (1 - \alpha) * x_2)). \qquad \triangleleft \quad \textit{pixel $\alpha$-blend} \qquad (8)$$

We select the weight $\alpha$ to be proportional to the area of the target modification. Qualitatively, the composition method is better able to change the target region while keeping the context region fixed, e.g., add white windows while reducing changes in the fireplace or couch in Fig. 7, whereas the other two $\alpha$-blending methods are less faithful in preserving image content. To quantify this effect, we compute the masked $L_1$ distance (Eqn. 5) of the interpolated images to (1) the original masked context region, (2) to the original masked target region, and (3) to the input collage over 500 random samples. The composition method obtains lower distance from the target and context, and is also closer to the desired collage. Unlike interpolations using attribute vectors, composition manipulations do not need to be learned from data - they are based on a single context and target image, and also allow multiple possible directions of manipulation. We show additional interpolations and comparison to learned attribute vectors in Supp. Sec. A.2.2, and additional applications such as dataset rebalancing and one-shot image editing in Supp. Sec. A.2.1.

### 4.4 USING REGRESSION TO INVESTIGATE INDEPENDENCE OF IMAGE COMPONENTS

Given these unsupervised objected representations, we seek to investigate the independence of individual components by minimizing "leakage" of the desired edits. For example, if we change a facial expression from a frown to a smile, the hair style should not change. We quantify the independence of parts under the global coherence constraint imposed by the regressor and generator pair by first parsing a given face image $x$ into separate semantic components (such as eyes, nose, etc.) represented as masks $m_c$. For each component, we generate $N$ new faces $x_n$, and replace the component region $m_c$ in $x$ by the corresponding region in $x_n$, yielding (after regression and generation) for each component $c$ a set of $N$ images $x_{c,n} = G(E(m_c \otimes x_n + (1 - m_c) \otimes x))$. We can now measure how much changing component $c$ changes each pixel location by computing the normalized pixel-wise standard deviation of $x_{c,n}$ across the $n$ different replacements as $v_c = \sigma_c / \sum_c \sigma_c$, where $\sigma_c = \sqrt{\mathbb{E}_n[(x_{n,c} - \mathbb{E}_n[x_{n,c}])^2]}$. For a given component $c$, we measure independence as the average

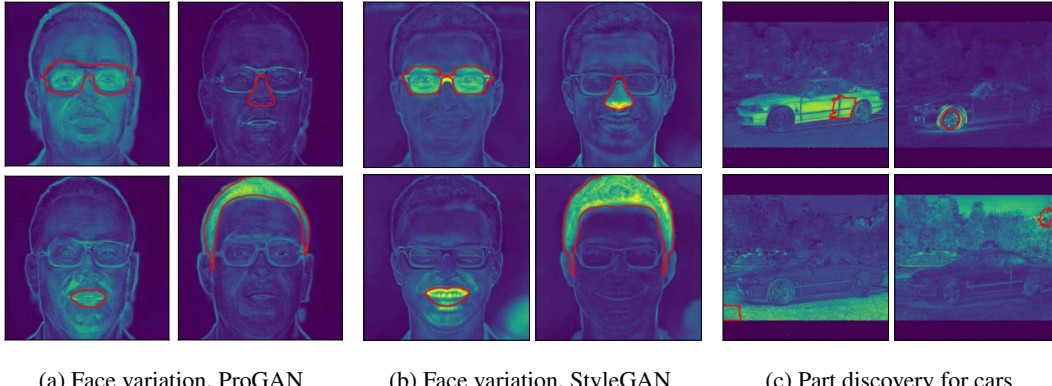

(a) Face variation, ProGAN     (b) Face variation, StyleGAN     (c) Part discovery for cars

Figure 8: Image variation when replacing single parts. In ProGAN (a), replacing single parts leads to change beyond the part that is being changed. Changes in StyleGAN (b) are visually more localized. This method can be used to find correlated regions even without semantic labels, shown for cars (c).

Table 1: Face part independence across models (lower means more independent), measured as the variation outside of the replaced part, computed over difference replacements.

|          | Background | Skin  | Eyes  | Ears  | Nose  | Mouth | Hair  | Average |
|----------|-----------|-------|-------|-------|-------|-------|-------|---------|
| ProGAN   | 0.167     | 0.177 | 0.043 | 0.024 | 0.030 | 0.038 | 0.170 | 0.093   |
| StyleGAN | 0.148     | 0.248 | 0.062 | 0.024 | 0.065 | 0.045 | 0.140 | 0.105   |

variation outside of $c$ that results from modifying $c$ as $s_c = \mathbb{E}[(1 - m_c) \otimes v_c]$ (a lower $s_c$ means higher independence). We repeat this experiment 100 times and use $N = 20$ samples.

Table 1 shows the variations of ProGAN and StyleGAN. StyleGAN better separates the background from the face and reduces leakage when changing the hair; for the face parts, leakage is small for both networks. A notable exception is the "skin" area, which for StyleGAN is more globally entangled. This might be because StyleGAN is generally better able to reason about global effects such as illumination, which are strongly reflected in the appearance of the skin, yet have a global impact on the image. Figure 7(a) and (b) qualitatively show examples for the variation maps $v_c$ for different parts for ProGAN (a) and StyleGAN (b); the replaced part is marked in red. Lastly, this method can be utilized for unsupervised part discovery, as shown in Fig. 7(c). Here, changing the color of the rear door (top left) changes the appearance of the whole car body; a change of the tire (top right) is very localized, and the foreground (bottom left) and background (bottom right) are large parts varying together, but distinct from the car. More examples of part variations are shown in Supp. Sec. A.2.5.

## 5    CONCLUSION

Using a simple latent space regression model, we investigate the compositional properties of pretrained GAN generators. We train a regressor model to predict the latent code given an input image with the additional objective of learning to complete a scene with missing pixels. With this regressor, we can probe various properties and biases that the GAN learns from data. We find that, in creating scenes, the GAN allows for local degrees of freedom but maintains an overall degree of global consistency; in particular, this compositional representation of local structures is already present at the level of the latent code. This allows us to input approximate templates to the regressor model, such as partially occluded scenes or collages extracted from parts of images, and use the regressor and generator as image prior to regenerate a realistic output. The latent regression approach allows us to investigate how the GAN enforces independence of image parts, while being trained without knowledge of objects or explicit attribute labels. It only requires a single forward pass on the models, which enables real time image editing based on single image examples.

ACKNOWLEDGMENTS

We would like to thank David Bau, Richard Zhang, Tongzhou Wang, and Luke Anderson for helpful discussions and feedback. Thanks to Antonio Torralba, Alyosha Efros, Richard Zhang, Jun-Yan Zhu, Wei-Chiu Ma, Minyoung Huh, and Yen-Chen Lin for permission to use their photographs in Fig. 23. LC is supported by the National Science Foundation Graduate Research Fellowship under Grant No. 1745302. JW is supported by a grant from Intel Corp.

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

# A APPENDIX

## A.1 SUPPLEMENTARY METHODS

### A.1.1 ADDITIONAL TRAINING DETAILS

The loss function of the encoder contains image loss terms to ensure that the output of the generator approximates the target image, and a latent recovery loss term to ensure that the predicted latent code matches the original latent code. On the image side, we use mean square error loss in conjunction with LPIPS perceptual loss (Zhang et al., 2018). The latent recovery loss depends on the type of GAN. Due to pixel normalization in ProGAN, we use a latent recovery loss based on cosine similarity, as the exact magnitude of the recovered latent code does not matter after normalization:

$$L_z = 1 - \frac{z}{||z||_2} \cdot \frac{E(x)}{||E(x)||_2}. \tag{9}$$

For StyleGAN, we invert to an intermediate latent space, as it is known that in this space semantic properties are better disentangled than in $\mathcal{Z}$ (Karras et al., 2019a). Furthermore, allowing the latents to differ on different scales has been shown to better capture the variability of real images (Abdal et al., 2019). During training, we therefore generate different latents for different scales, and train the encoder to estimate different styles, i.e. estimate $w \in \mathcal{W}^+$. Unlike the latent space of ProGAN, however, $w \in \mathcal{W}^+$ is not normalized to the hypersphere. Instead of a cosine loss, we therefore use a mean square error loss as the latent recovery loss:

$$L_w = ||w - E(x)||_2. \tag{10}$$

We train the encoders using a ResNet backbone (ResNet-18 for ProGAN, and ResNet-34 for Stylegan; He et al. (2016)), modifying the output dimensionality to match the number of latent dimensions for each GAN. The encoders are trained with the Adam optimizer (Kingma & Ba, 2014) with learning rate $lr = 0.0001$. Training takes from two days to about a week on a single GPU, depending on the resolution of the GAN. For ProGAN encoders, we use batch size 16 for the 256 resolution generators, and train for 500K batches. For the 1024 resolution generator, we use batch size 4 and 400K batches. We train the StyleGAN encoders for 680k batches (256 and 512 resolution) or 580k batches (1024 resolution), and add identity loss (Richardson et al., 2020) with weight $\lambda = 1.0$ on the FFHQ encoder.

When training with masks, we take a small 6x6 patch of random uniform noise $u$, upsample to the generator's resolution, and sample a threshold $t$ from the uniform distribution in range $[0.3, 1.0]$ to create the binary mask:

$$m = \mathbb{1}[\text{Upsample}(u) > t]$$
$$x_m = m \otimes x \tag{11}$$

We also experimented with masks comprised of random rectangular patches (Zhang et al., 2017), but obtained qualitatively similar results. At inference time, the exact shape of the mask does not matter: we can use hard coded rectangles, hand-drawn masks, or masks based on other pretrained networks. Note that the mask does not distinguish between input image parts – it is a binary mask with value 1 where the generator should try to reconstruct, and 0 where the generator should fill in the missing region.

### A.1.2 ADDITIONAL DETAILS ON COMPOSITION

When creating automated collages from image parts, we use a pretrained segmentation network (Xiao et al., 2018) to extract parts from randomly sampled individual images. We manually define a set of segmentation class for a given image class, and, to handle overlap between parts, specify an order to these segmentation classes. To generate a collage, we then sample one random image per class. For church scenes, we use an ordering of (from back to front) sky, building, tree, and foreground layers – this ensures that a randomly sampled building patch will appear in front of the randomly sampled sky patch. For living rooms, the ordering we use is floor, ceiling, wall, painting, window, fireplace, sofa, and coffee table – again ensuring that the more foreground elements are layered on top. For cars, we use sky, building, tree, foreground, and car. For faces we order by background, skin, eye, mouth, nose, and hair. In Sec. A.2.4 we investigate using other methods to extract patches from images, rather than image parts derived from segmentation classes.

## A.2 Supplementary Results

### A.2.1 Additional Applications

In the main text, we primarily focus on using the regression network as a tool to understand how the generator composes scenes from missing regions and disparate image parts. However, because the regressor allows for fast inference, it enables a number of other real-time image synthesis applications.

**Image Completion From Masked Inputs.** Using the masked latent space regressor in Eqn. 3, we investigate the GAN's ability to automatically fill in unknown portions of a scene given incomplete context. These reconstructions are done on parts of real images to ensure that the regressor is not simply memorizing the input, as could be the case with a generated input (the regressor is never trained on real images). For example, when a headless horse is shown to the masked regressor, the rest of the horse can be filled in by the GAN. In contrast, a regressor that is unaware of missing pixels (Eqn. 2; RGB) is unable to produce a realistic output. We show qualitative examples in Fig. 9.

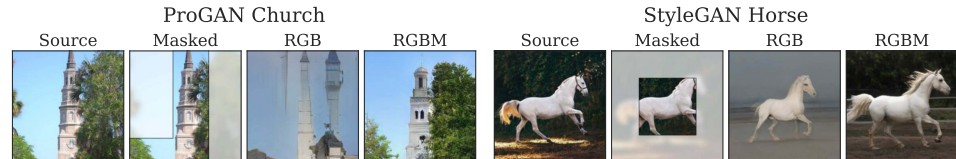

Figure 9: Given a masked real image, a regressor without knowledge of masked images (RGB) is unable to realistically reconstruct the scene, while the regressor trained on masks inputs inpaints in the unknown region in a way that is consistent with the given context (RGBM).

**Multimodal Editing.** Because the regressor only requires a single example of the property we want to reconstruct, it is possible to achieve multimodal manipulations simply by overlaying different desired properties on a given context image. Here, we demonstrate an example of adding different styles of trees to a church scene. In each of the context images, there is originally no tree on the right-hand side. We can add different types of trees to each context image simply by replacing some pixels with tree pixels from a different image, and performing image inversion to create the composite. Here, we use a rectangular region as a mask, rather than following the boundary of the tree precisely. However, note that, after inversion, the color of the sky remains consistent in the composite image, and so does the building color in second tree example. This image editing approach does not require learning separate clusters or having labelled attributes, and therefore can be done with a single pair of images without known cluster definitions, unlike the methods of Bau et al. (2018) and Collins et al. (2020). Furthermore, unlike methods based on segmentation maps (Park et al., 2019; Gu et al., 2019b), styles within each individual semantic category, e.g. the color of the sky, is also changeable based on the provided input to the encoder.

**Attribute Editing With A Single Unlabeled Example.** Typically in attribute editing using generative models, one requires a collection of labeled examples of the attribute to modify; for example, we take a collection of generated images with and without the attribute based on an attribute classifier, and find the average difference in their latent codes (Jahanian et al., 2020; Radford et al., 2015; Goetschalckx et al., 2019; Shen et al., 2019). Here, we demonstrate an approach towards attribute editing without using attribute labels (Fig. 11). We overlay a single example of our target attribute (e.g. a smiling mouth or a round church tower), on top of the context image we want to modify, and encode it to get the corresponding latent code. We then subtract the difference between the original and modified latent codes, and add it to the latent code of the secondary context image: $z_{1,\text{modified}} - z_1 + z_2$. This bypasses the need for an attribute classifier, such as one that identifies round or square church towers.

**Dataset Rebalancing.** Because the latent regression inverter only requires a forward pass through the network, we can quickly generate a large number of reconstructions. Here, we use this property to investigate rebalancing a dataset (Fig. 12). Using pretrained attribute detectors from Karras et al. (2019a), we first note that there is a smaller fraction of smiling males than smiling females in CelebA-HQ (first panel), although the detector itself is biased and tends to overestimate smiling in general

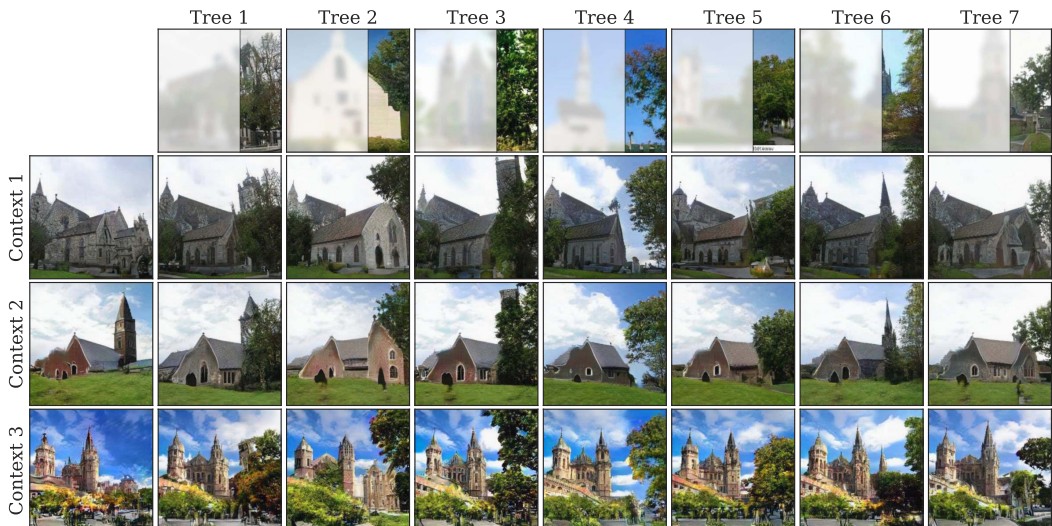

Figure 10: Using latent regression, we can perform multimodal editing to a context image. Each editing operation uses a single pair of images. Here, our editing mask is a rough rectangular box. Note how in case of *Tree 2*, the pasted region includes part of the church; the regressor attempts to include this part, and the resulting re-generated building therefore looks different from the building in the other examples.

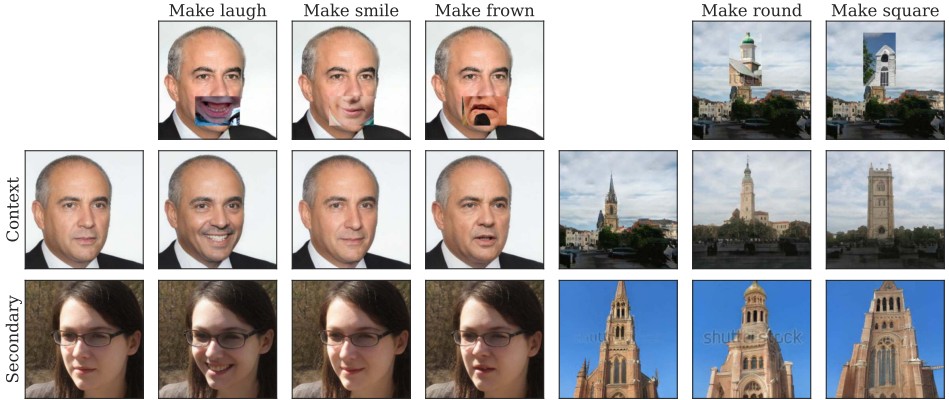

Figure 11: We demonstrate the transfer capability of the latent code manipulations. From a single pair of images, we can compute a manipulation vector and apply it to a secondary image, which edits the secondary image accordingly.

compared to ground truth labels (second panel). The detections in the GAN generated images mimic this imbalance (third panel). Next, using a fixed set of generated images, we randomly swap the mouth regions among them in accordance to our desired proportion of smiling rates – we give each face a batch of 16 possible swaps and taking the one that yields the strongest detection as smiling/not smiling based on our desired characteristic (if no swaps are successful, the face is just used as is). We use a hardcoded rectangular box region around the mouth, and encode the image through the generator to blend the modified mouth to the face. After performing swapping on generated images, this allows us rebalance the smiling rates, better equalizing smiling among males and females (fourth panel). Finally, we use the latent regression to perform mouth swapping on the dataset, and train a secondary GAN on this modified dataset – this also improves the smiling imbalance, although the effect is stronger in males than females (fifth panel). We note that a limitation of this method is that the rebalanced proportion uses the attribute detector as supervision, and therefore a biased or inaccurate attribute detector will still result in biases in the rebalanced dataset.

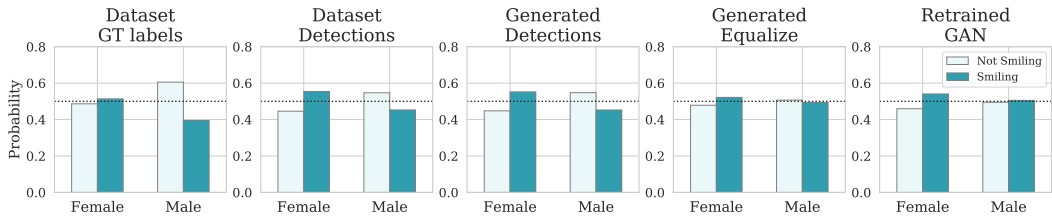

Figure 12: We perform dataset rebalancing using latent space regression. From pretrained attribute detectors, a smaller fraction of males smile than females in CelebA-HQ, and the GAN samples mimic this trend. By swapping mouths in the GAN samples, we can equalize male and female smiling rates. Next, we do this swap and invert operation on the dataset and retrain a GAN on the modified dataset, which also improves balance in smiling rates although the effect is stronger for the male category.

### A.2.2 COMPARING COMPOSITION WITH LATENT SPACE INTERPOLATION

In the main text, we compare our composition approach to two types of interpolations – latent $\alpha$-blending and pixel $\alpha$-blending – on a living room generator. Here, we show equivalent examples in church scenes. In Fig. 13, we demonstrate a "tree" edit, in which we want the background church to remain the same but trees to be added in the foreground. Compared to latent and pixel $\alpha$-blending, the composition approach better preserves the church while adding trees, which we quantify using masked L1 distance. Similarly, we can change the sky of context scene – e.g. by turning a clear sky into a cloudy one in Fig. 14, where again, using composition better preserves the details of the church as the sky changes, compared to $\alpha$-blending.

In Fig 15, we show the result of changing the smile on a generated face image. For the smile attribute, we also compare to a learned smile edit vector using labelled images from a pretrained smile classifier. We additionally measure the the facial embedding distance using a pretrained face-identification network[2] – the goal of the interpolation is to change the smile of the image while reducing changes to the rest of the face identity, and thus minimize embedding distance. Because the mouth region is small, choosing the interpolation weight $\alpha$ by the target area minimally changes the interpolated image ($\alpha > 0.99$), so instead we use an $\alpha = 0.7$ weight so that all methods have similar distance to the target. While the composition approach and applying learned attribute vector perform similarly, learning the attribute vector requires labelled examples, and cannot perform multimodal modifications such as applying different smiles to a given face.

### A.2.3 LOSS ABLATIONS

When training the regression network, we use a combination of three losses: pixel loss, perceptual loss (Zhang et al., 2018), and a latent recovery loss. In this section, we investigate the reconstruction result using different combinations of losses on the ProGAN church generator. In the first case, we do not enforce latent recovery, so the encoded latent code does not have to be close to the ground truth latent code but the encoder and generator just need to reconstruct the masked input image. In the second case, we investigate omitting the perceptual loss, and simply use an L2 loss in image space. Since the encoders are trained with L2 loss and the VGG variant of perceptual loss, we evaluate with L1 reconstruction and the AlexNet variant of perceptual loss. Training with all losses leads to reconstructions that are more perceptually similar (lower LPIPS loss) compared to the two other variants, while per-pixel L1 reconstruction is not greatly affected (Tab. 2). We show qualitative examples of the reconstructions on masked input in Fig. 16.

Table 2: Table of masked L1 and LPIPS-Alexnet reconstruction errors for encoder networks trained with all losses, no latent loss, and no perceptual loss.

|  | All Losses | No Latent | No LPIPS |
|---|---|---|---|
| L1 | 0.194 | 0.194 | 0.197 |
| LPIPS(alex) | 0.291 | 0.305 | 0.318 |

---

[2] https://github.com/timesler/facenet-pytorch

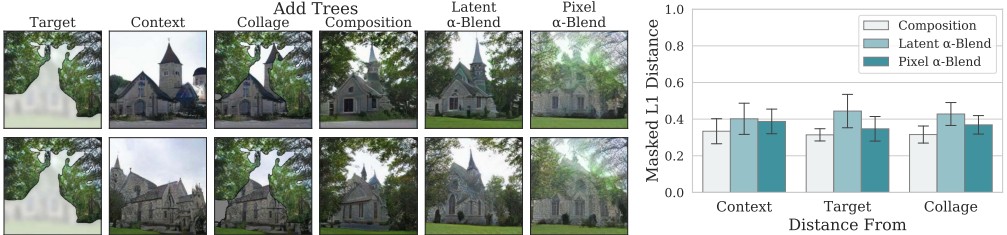

Figure 13: Comparison of image editing outcomes when adding trees (Target) to different scenes (Context). The overlay of these two components is shown in Collage, where we use a mask to indicate missing regions. We compare (1) Composition: using the encoder and generator to blend the images together to (2) Latent $\alpha$-blend: interpolation in the latent codes of the encoded images and (3) Pixel $\alpha$-blend: interpolation in pixel space, which then passes through the encoder to pull the image closer to the image manifold. (Right) Over 500 randomly sampled images, we find that the composition approach is better able to match properties of both the context and target images compared to the two alpha-blending techniques.

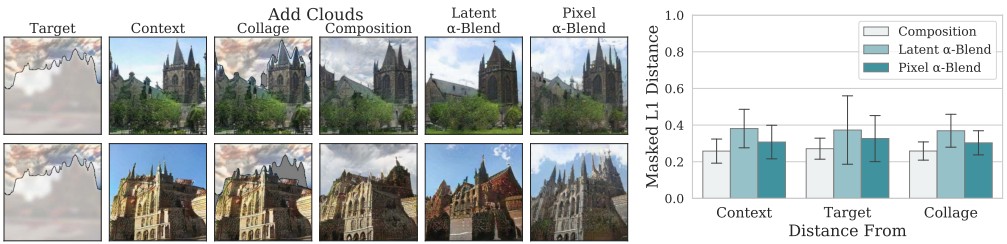

Figure 14: Comparison of image editing outcomes when changing the sky of a scene, using similar encoder-based Composition, Latent $\alpha$-blend, and Pixel $\alpha$-blend methods as above.

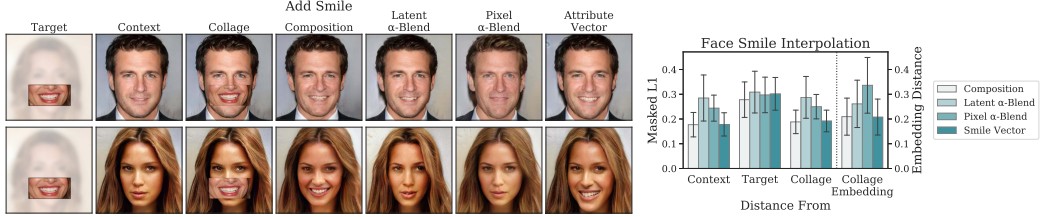

Figure 15: Comparison of image editing outcomes using the same encoder-based Composition, Latent $\alpha$-blend, and Pixel $\alpha$-blend methods as above when adding a smile to a face. We also compare these approaches to the modifications produced by edit vector learned from attribute labels.

### A.2.4 ADDITIONAL COMPOSITION RESULTS

**Comparing different composition approaches.** In Fig 17, we show examples of extracted image parts, the collage formed by overlaying the image parts, and the result of poisson blending the images according to the outline of each extracted part. We further compare to variations of the encoder setup, where (1) RGB: the encoder is not aware of missing pixels in the collaged input, (2) RGB Fill: we fill the missing pixels with the average color of the background layer, and (3) RGBM: we allow the encoder and generator to inpaint on missing regions. Table 3 shows image quality metrics of FID (lower is better; Heusel et al. (2017)) and density and coverage (higher is better; Naeem et al. (2020)) and masked L1 reconstruction (lower is better) for each generator and domain. To obtain feature representations for the density and coverage metrics, we resize all images to 256px and use pretrained VGG features prior to the final classification layer (Simonyan & Zisserman, 2015). While the composite input collages are highly unrealistic with high FID and low density/coverage, the inverted images are closer to the image manifold. Of the three inversion methods, the RGBM inversion tends to yield lower FID and higher density and coverage, while minimizing L1 reconstruction error. We compare the GAN inversion methods to Poisson blending (Pérez et al., 2003), in which we incrementally blend the 4-8 image layers on top of each other using their respective masks. As

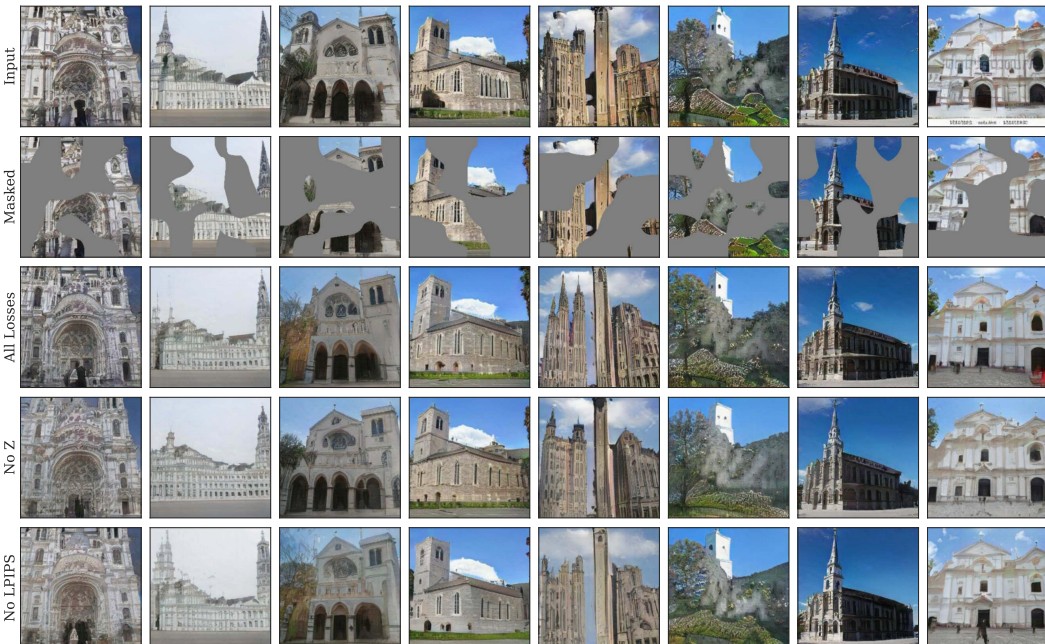

Figure 16: Qualitative examples of reconstructions from masked inputs on encoders trained with different combinations of loss terms.

Poisson blending does not naturally handle inpainting, we do not mask the bottom-most image layer, but rather use it to fill in any remaining holes in the blended image. We find that Poisson blending is unable to create realistic composites, due to the several overlapping, yet misaligned, image layers used to create the composites.

Table 3: Quantitative comparison of automated collaging and latent regression variations. The latent regressor and generator pulls the unrealistic composite images closer to the real manifold, yielding high density and coverage and lower FID on output, compared to the collaged inputs and Poisson blending. For ProGAN models, we use PyTorch models from Bau et al. (2019); for StyleGAN models, we use a Pytorch conversion of the Tensorflow models from Karras et al. (2019a).

| Model | Metric | GAN Samples | GAN Reconstructions | Collage | Collage Filled | Poisson Blended | RGB Inverted | RGB Filled Inverted | RGBM Inverted |
|---|---|---|---|---|---|---|---|---|---|
| ProGAN Church | FID | 8.01 | 6.72 | 21.20 | 27.35 | 24.12 | 10.85 | 12.90 | 9.40 |
| | Density | 0.94 | 1.04 | 0.15 | 0.15 | 0.45 | 1.02 | 0.92 | 1.23 |
| | Coverage | 0.78 | 0.82 | 0.22 | 0.22 | 0.41 | 0.71 | 0.74 | 0.78 |
| | Masked L1 | – | – | – | – | – | 0.26 | 0.28 | 0.26 |
| ProGAN Living Room | FID | 13.17 | 10.46 | 72.82 | 69.95 | 47.84 | 19.50 | 17.70 | 14.90 |
| | Density | 0.69 | 0.86 | 0.01 | 0.03 | 0.28 | 0.98 | 0.81 | 1.05 |
| | Coverage | 0.63 | 0.69 | 0.02 | 0.02 | 0.26 | 0.60 | 0.59 | 0.64 |
| | Masked L1 | – | – | – | – | – | 0.33 | 0.34 | 0.30 |
| ProGAN CelebA-HQ | FID | 10.43 | 12.38 | 81.82 | 82.09 | 53.76 | 16.24 | 15.35 | 17.41 |
| | Density | 1.27 | 1.55 | 0.11 | 0.14 | 0.28 | 1.14 | 1.19 | 1.69 |
| | Coverage | 0.83 | 0.84 | 0.03 | 0.03 | 0.15 | 0.72 | 0.74 | 0.78 |
| | Masked L1 | – | – | – | – | – | 0.26 | 0.26 | 0.24 |
| StyleGAN Church | FID | 3.90 | 4.27 | 15.66 | 22.89 | 16.92 | 6.13 | 6.75 | 6.09 |
| | Density | 0.88 | 1.06 | 0.15 | 0.14 | 0.41 | 1.31 | 1.31 | 1.50 |
| | Coverage | 0.85 | 0.87 | 0.26 | 0.25 | 0.50 | 0.85 | 0.86 | 0.87 |
| | Masked L1 | – | – | – | – | – | 0.32 | 0.32 | 0.30 |
| StyleGAN Car | FID | 2.31 | 2.98 | 17.44 | 141.21 | 15.38 | 6.23 | 5.87 | 5.38 |
| | Density | 1.07 | 1.32 | 0.37 | 0.42 | 0.61 | 1.69 | 1.53 | 1.51 |
| | Coverage | 0.94 | 0.94 | 0.44 | 0.36 | 0.54 | 0.92 | 0.91 | 0.91 |
| | Masked L1 | – | – | – | – | – | 0.32 | 0.34 | 0.30 |
| StyleGAN FFHQ | FID | 2.86 | 6.27 | 92.15 | 92.00 | 36.54 | 26.25 | 25.63 | 24.09 |
| | Density | 1.17 | 1.61 | 0.16 | 0.17 | 0.26 | 1.67 | 1.58 | 2.01 |
| | Coverage | 0.90 | 0.89 | 0.02 | 0.02 | 0.24 | 0.67 | 0.67 | 0.74 |
| | Masked L1 | – | – | – | – | – | 0.30 | 0.30 | 0.28 |

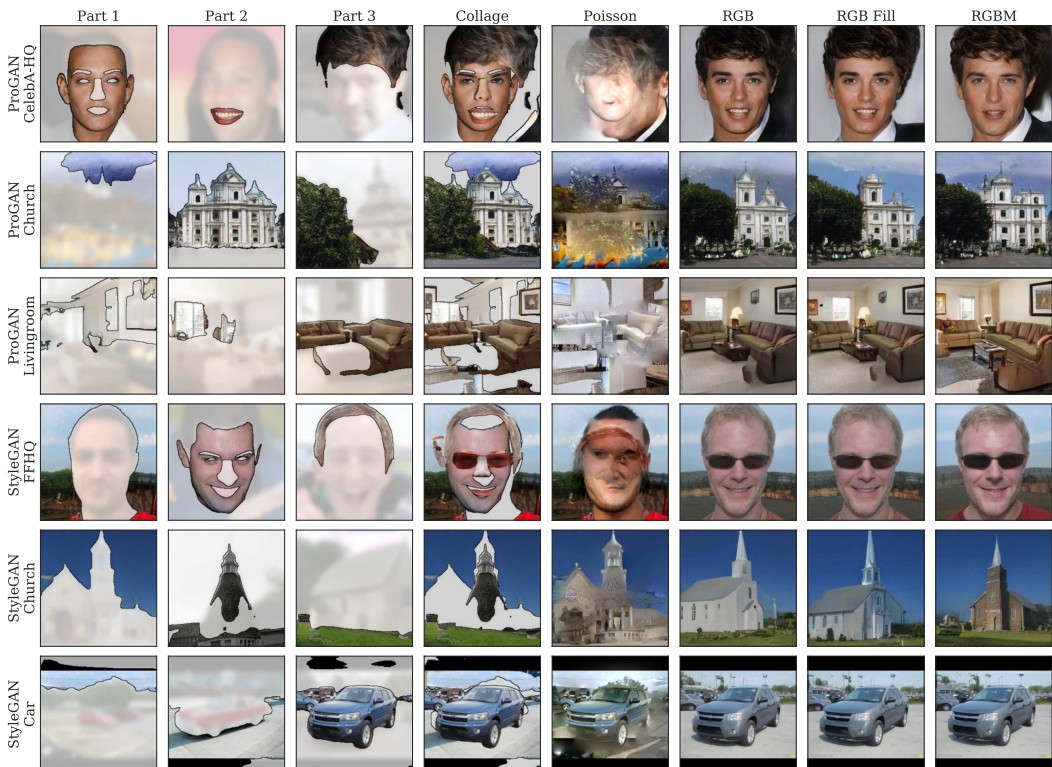

Figure 17: Qualitative examples of automatic image composition. We extract parts of sampled images and overlay them to form a rough collage. We compare Poisson blending the image parts (we use the bottom-most image layer to fill in any remaining holes in the collaged image) to encoder-based methods that invert the collage through the generator, either without knowledge of missing pixels (RGB and RGB Fill methods), or with the objective of inpainting the missing regions (RGBM). While the RGB and RGB Fill methods also demonstrate an alignment-correcting effect, we find quantitatively that the RGBM encoder tends to have lower FID over 50K samples (Tab. 3).

**Comparing different image reconstruction generators.** How are image priors different across different image reconstruction pipelines? Our encoder method relies on a pretrained generator, and trains an encoder network to predict the generator's latent code of a given image. Therefore, it can take advantage of the image priors learned by the generator to keep the result close to the image manifold. Here, we compare to different image reconstruction approaches, such as autoencoder architectures or optimization methods rather than feed-forward inference. We construct the same set of input collages using parts of real images and compare a variety of image reconstruction methods encompassing feed-forward networks, pretrained GAN models, encoder networks. Since some reconstruction methods are optimization-based and thus take several minutes, we use a set of 200 images. We then compute the L1 reconstruction error in the valid region of the input, density (measures proximity to the real image manifold; Naeem et al. (2020)), and FID (measures realism; Heusel et al. (2017); but note that we are limited to small sample sizes due to optimization time).

For the church domain, we first compare to four methods that do not rely on a pretrained GAN network; rather, the generator and encoder is jointly learned using an autoencoder-like architecture. (1) We train a CycleGAN (Zhu et al., 2017) between image collages and the real image domain, creating a network that is explicitly trained for image composition in an unpaired manner, as there are no ground-truth "composited" images for the randomly generated image collages. (2) We use a pretrained SPADE (Park et al., 2019) network which creates images from segmentation maps, but information about object style (e.g. color and texture) is lost in the segmentation input. (3) We use a pretrained inpainting model that is trained to fill in small holes in the image (Yu et al., 2018), but does not correct for misalignments or global color inconsistencies between image parts. (4) We train Deep Image Prior (DIP) networks (Ulyanov et al., 2018) which performs test-time optimization on an autoencoder to reconstruct a single image, where using a masked loss allows it to inpaint missing regions.

Next, we use the ProGAN and StyleGAN2 pretrained generators, and experiment with different ways of inverting into the latent code. Methods that leverage a pretrained GAN for inversion, but are optimization-based rather than feed-forward include (5&6) LBFGS methods (Liu & Nocedal, 1989) on ProGAN and StyleGAN, which iteratively optimizes for the best latent code starting from the best latent among 500 random samples, (7) Multi-Code Prior (Gu et al., 2019a), which combines multiple latent codes in the intermediate layers of the GAN, and (8) a StyleGAN2 projection method using perceptual loss (Karras et al., 2019a). For all optimization-based GAN inversion methods, we modify the objective with a masked loss to only reconstruct valid regions of the input.

We use our trained regressor network for the remaining comparisons. (9&10) We use our ProGAN and StyleGAN regressors to encode the input image as initialization, and then perform LBFGS optimization. (11&12) We use our ProGAN and StyleGAN regressors in a fully feed-forward manner.

Tab. 4 summarizes the methods and illustrates the tradeoff between reconstruction (L1), realism (Density and FID), and optimization time. Due to the unrealistic nature of the input collages, a method that reduces reconstruction error is less realistic, whereas a more realistic output may offer a worse reconstruction of the input. Furthermore, methods that are not feed-forward incur substantially more time per image. Fig 18 shows qualitative results, where in particular the third example is a challenging input where the lower part of the tower is missing. The two encoders demonstrate an image prior in which the bottom of the tower is reconstructed on output. While DIP can inpaint missing regions, it cannot leverage learned image prior. CycleGAN can fill in missing patterns, but with visible artifacts, whereas SPADE changes the style of each input component. Iteratively optimizing on a pretrained generator can lose semantic priors as it optimizes towards an unrealistic input.

Table 4: On a set of 200 input collages constructed from random image parts, we compare compositional properties of several image reconstruction approaches including methods based on autoencoder-style networks that do not leverage a pretrained GAN, methods based on optimization on the latent code of a pretrained GAN, and methods based on encoder networks to regress the latent code. Autoencoder-based and optimization-based methods can achieve lower reconstruction error (L1; lower is better), but are less realistic (Density; higher is better). Another tradeoff is optimization time, as feed-forward methods are orders of magnitude faster than optimization-based approaches. We also report FID (lower is better), but our sample size is limited due to the latency of optimization-based approaches.

| Inversion Method | GAN-Based | Encoder-Based | Feed-Forward | L1 ($\downarrow$) | Density ($\uparrow$) | FID ($\downarrow$) | Time (s) ($\downarrow$) |
|---|---|---|---|---|---|---|---|
| 1. CycleGAN | | | ✓ | 0.14 | 0.43 | 74.63 | 0.10 |
| 2. SPADE | | | ✓ | 0.50 | 1.01 | 137.96 | 0.07 |
| 3. Inpainting | | | ✓ | 0.00 | 0.34 | 72.86 | 0.02 |
| 4. DIP | | | | 0.04 | 0.18 | 86.44 | 98.90 |
| 5. ProGAN LBFGS | ✓ | | | 0.23 | 1.04 | 53.64 | 188.94 |
| 6. StyleGAN LBFGS | ✓ | | | 0.13 | 0.50 | 84.97 | 368.92 |
| 7. Multi-Code Prior | ✓ | | | 0.13 | 0.21 | 104.34 | 477.93 |
| 8. StyleGAN Projection | ✓ | | | 0.23 | 0.25 | 62.62 | 86.87 |
| 9. ProGAN Encode+LBFGS | ✓ | ✓ | | 0.23 | 0.88 | 49.60 | 66.55 |
| 10. StyleGAN Encode+LBFGS | ✓ | ✓ | | 0.15 | 0.38 | 64.29 | 176.96 |
| 11. Ours: ProGAN Encoder | ✓ | ✓ | ✓ | 0.32 | 0.79 | 55.11 | 0.02 |
| 12. Ours: StyleGAN Encoder | ✓ | ✓ | ✓ | 0.36 | 1.86 | 48.08 | 0.03 |

On the face domain, we compare (1) the inpainting method of Yu et al. (2018) pretrained on faces, (2) the Im2StyleGAN method (Abdal et al., 2019) which optimizes within the $\mathcal{W}+$ latent space of StyleGAN, and (3) the ALAE model (Pidhorskyi et al., 2020) which jointly trained the encoder and generator. More similar to our approach, (4&5) the In-domain encoding and diffusion methods (Zhu et al., 2020) encodes and optimizes for a latent code of StyleGAN that matches the target image. (6) We modify and retrain the Pixel2Style2Pixel network (PSP; Richardson et al. (2020)), which also leverages the StyleGAN generator, to perform regression with arbitrarily masked regions. The PSP network uses a feature pyramid to predict latent codes. (7&8) We use our regressor network on ProGAN and StyleGAN, which uses a ResNet backbone and predicts the latent code after pooling all spatial features.

Qualitatively, we find that the optimization-based Im2StyleGAN (Abdal et al., 2019) algorithm is not able to realistically inpaint missing regions in the input collage. While the ALAE autoencoder (Pidhorskyi et al., 2020) exhibits characteristics of blending the collage into a cohesive output image, the reconstruction error is higher than that of the GAN-based approaches. The In-domain encoder

method (Zhu et al., 2020) does not correct for misalignments in the inputs, resulting in low density, although the subsequent optimization step is able to further reduce L1 distance (Fig. 19). The PSP network modified with the masking objective is conceptually similar to our regressor; we find that it is better able to reconstruct the input image, but produces less realistic output. This suggests that an encoder which processes the input image globally before predicting the latent code output can help retain realism in output images. We measure reconstruction (L1) and realism (Density and FID) over 200 samples in Tab. 5.

Table 5: We construct 200 input collages from random face image parts, and compare to several image reconstruction methods. Again, we find a tradeoff between better reconstruction (low L1) and better realism (higher Density, lower FID), due to the imperfect nature of the input collages. There is no ground truth image that perfectly reconstructs the input yet is realistic.

| Inversion Method | GAN-Based | Encoder-Based | Feed-Forward | L1 ($\downarrow$) | Density ($\uparrow$) | FID ($\downarrow$) |
|---|---|---|---|---|---|---|
| 1. Inpainting | | | ✓ | 0.02 | 0.33 | 90.17 |
| 2. Im2StyleGAN | ✓ | | | 0.21 | 0.20 | 106.79 |
| 3. ALAE | | | ✓ | 0.33 | 0.47 | 80.13 |
| 4. In-domain Encoder | ✓ | ✓ | ✓ | 0.21 | 0.44 | 77.43 |
| 5. In-domain Diffusion | ✓ | ✓ | | 0.12 | 0.40 | 73.74 |
| 6. Masked PSP | ✓ | ✓ | ✓ | 0.15 | 0.52 | 73.06 |
| 7. Ours: ProGAN Encoder | ✓ | ✓ | ✓ | 0.27 | 1.55 | 57.20 |
| 8. Ours: StyleGAN Encoder | ✓ | ✓ | ✓ | 0.26 | 1.21 | 64.33 |

**Composing images using alternative definitions of image parts.**    In the main text, we focus on creating composite images using a pretrained segmentation network. However, we note that the exact process of extracting image parts does not matter, as the encoder and generator form an image prior that will remove inconsistencies in the input. In Fig. 20 we show composites generated using the output of a pretrained saliency network (Liu et al., 2018), and in Fig. 21 we show compositions using hand-selected compositional parts, where the parts extracted from each image do not have to correspond precisely with object boundaries.

**Editing with global illumination consistency.**    A property of the regressor network is that it enforces global coherence of the output, despite an unrealistic input, by learning a mapping from image to latent domains that is averaged over many samples. Thus, it is unable to perform exact reconstructions of the input image, but rather exhibits error-correcting properties when the input is imprecise, e.g. in the case of unaligned parts or abrupt changes in color. In Fig. 22, we investigate the ability of the regressor network to perform global adjustments to accommodate a desired change in lighting – such as adding a reflections or changing illumination *outside of the manipulated region* – to maintain realism at the tradeoff of higher reconstruction error.

**Improving compositions on real images.**    As the regressor network is trained to minimize reconstruction on average over all images, it can cause slight distortions on any given input image. To retain the compositionality effect of the regressor network, yet better fit a *specific* input image, we can finetune the weights of the regressor towards the given input image. Generally, a few seconds of finetuning suffices (30 optimizer steps; $< 5$ seconds), and subsequent editing on the image only requires a forward pass. We demonstrate this application in Fig. 23.

**Random composition samples.**    We show additional random samples of the generated composites across the ProGAN and StyleGAN2 generators for a variety of image domains in Fig. 24 and Fig. 25.

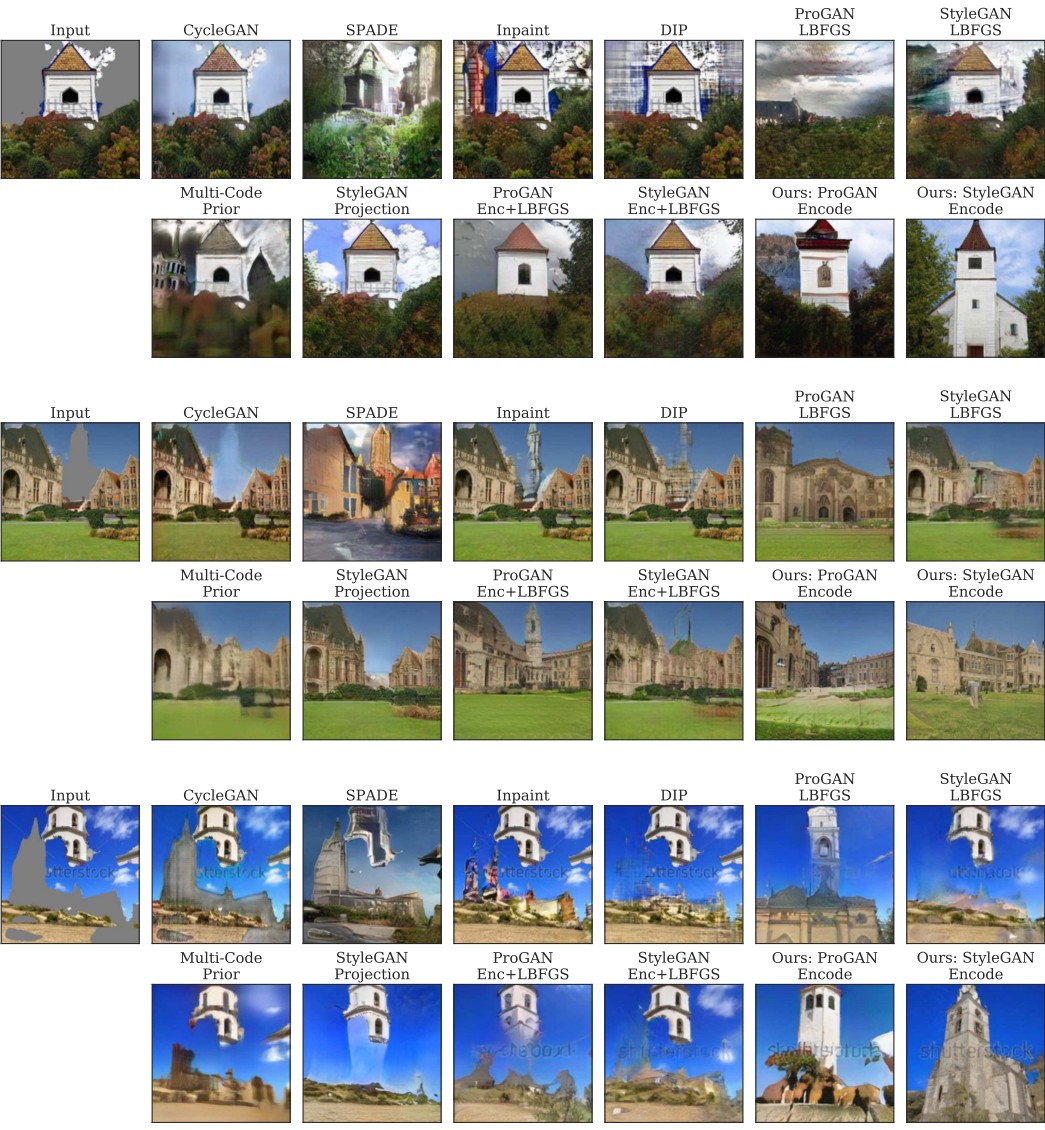

Figure 18: Qualitative examples of image composition comparing different encoder-decoder or generator-only setups. For each example, we show the input collage created from randomly selected parts of images (Input), and four autoencoder-based methods (CycleGAN, SPADE, Inpaint, DIP) in which the encoder and decode is trained jointly; as these are fully convolutional they lack global semantic priors. Next, we show four optimization-based methods (ProGAN & StyleGAN LBFGS, Multi-Code Prior, StyleGAN Projection); these can better match the target collage after optimizing for reconstruction, but they are orders of magnitude slower, making real-time interaction infeasible, and also lose semantic priors (as in the "floating tower" in the third example). We then show the combination of our encoder with LBFGS optimization, and our encoder using only a forward pass. While the feed-forward approach retains the tower prior and enables real-time interaction, its can distort the input image parts, illustrating a tradeoff between reconstruction and realism.

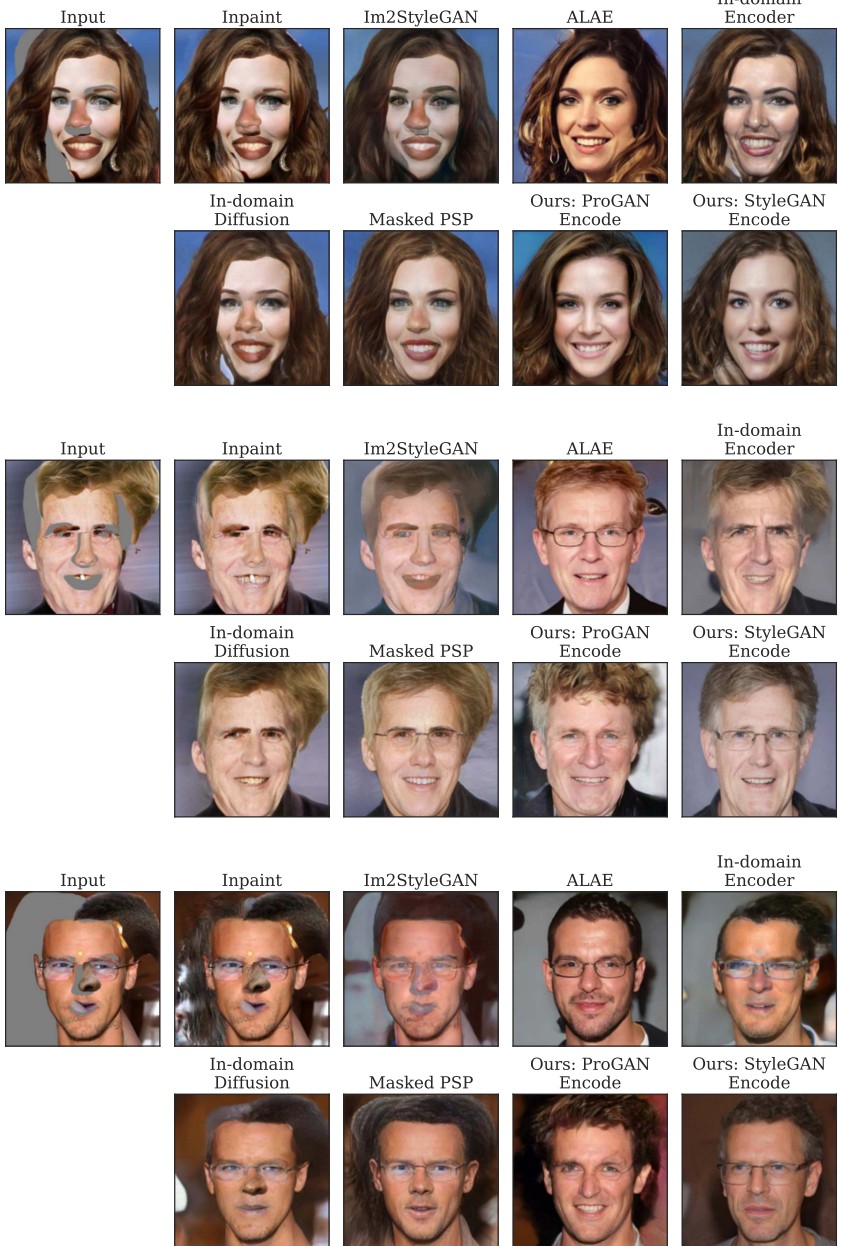

Figure 19: Qualitative examples of face composites across GAN-based generators. We show the input image, a collage from parts of random images (Input). We compare to Inpainting, Im2StyleGAN which iteratively optimizes for a latent code, the ALAE autoencoder where the generator and encoder is trained jointly, the In-domain encoder and diffusion which encodes and optimizes the latent, a masked version of the Pixel2Style2Pixel (PSP) network, and our masked encoder approaches (ProGAN Encoder and StyleGAN encoder).

ProGAN CelebA-HQ                                                    StyleGAN Car

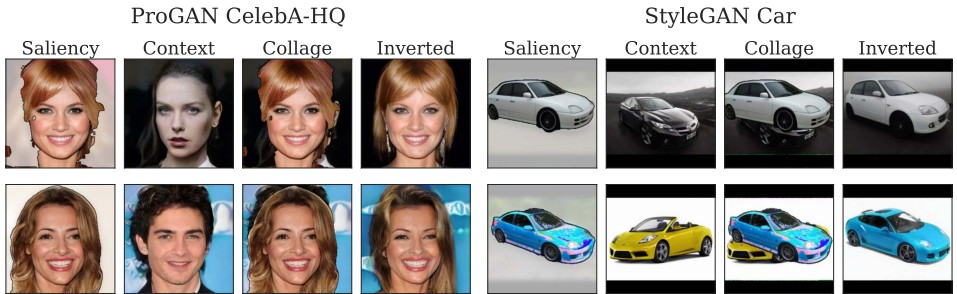

Figure 20: Qualitative examples of automatic image composition using a pretrained saliency network to generate input collages rather than a segmentation network. Note that in the second car example, the collage overlays the blue car with the yellow car still visible, but inversion via the encoder corrects this inconsistency.

ProGAN Living Room                                                  StyleGAN Church

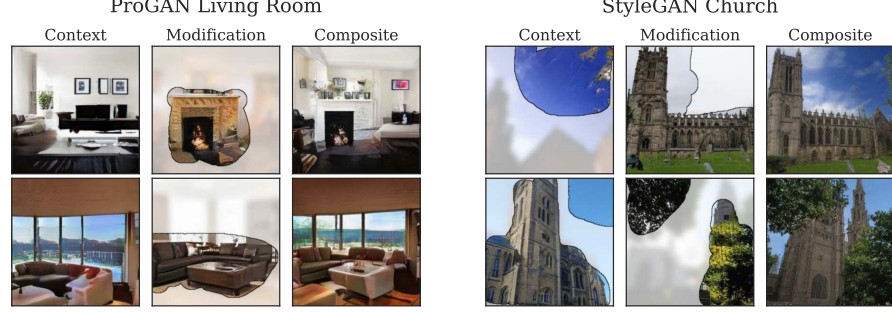

Figure 21: Qualitative examples of collages based on user-selected regions and the corresponding generator output. Note that the selected regions do not have to coincide neatly with object boundaries, and the generator will blend inconsistencies in the inputs together.

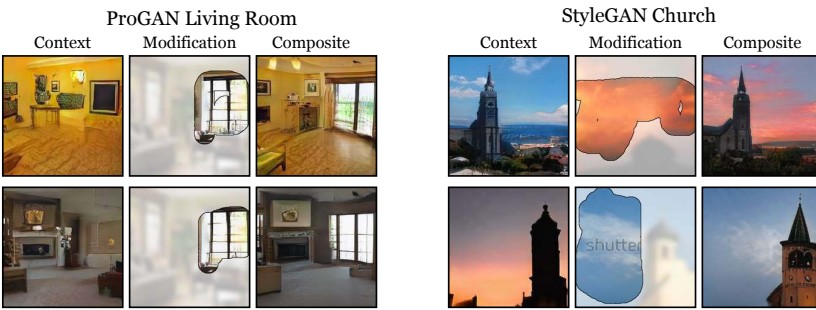

Figure 22: We investigate the capabilities of our instance-based regression and editing approach to modify lighting effects, similar to Bau et al. (2020). Editing lighting induces non-local changes, as the network must also change regions outside of the modified region to ensure global consistency – for example the reflection of the window on the floor (ProGAN Living Room), or changes in the illumination of the building (StyleGAN Church) – indicating a tradeoff between reconstructing the input versus creating a realistic image overall.

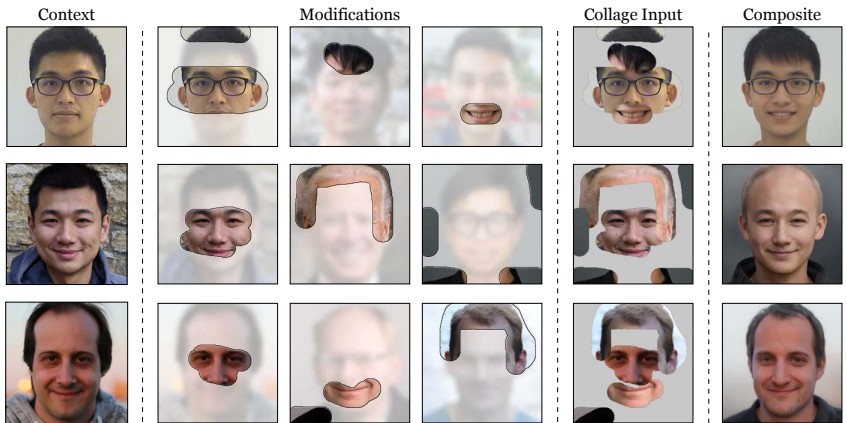

Figure 23: To improve the reconstruction towards real faces, yet preserve the properties of real-time composition, we can slightly finetune the regressor towards a context image (30 gradient steps, $< 5$ seconds). Subsequent modifications to the context image can be performed using just a feed-forward pass, enabling fast editing to create image composites.

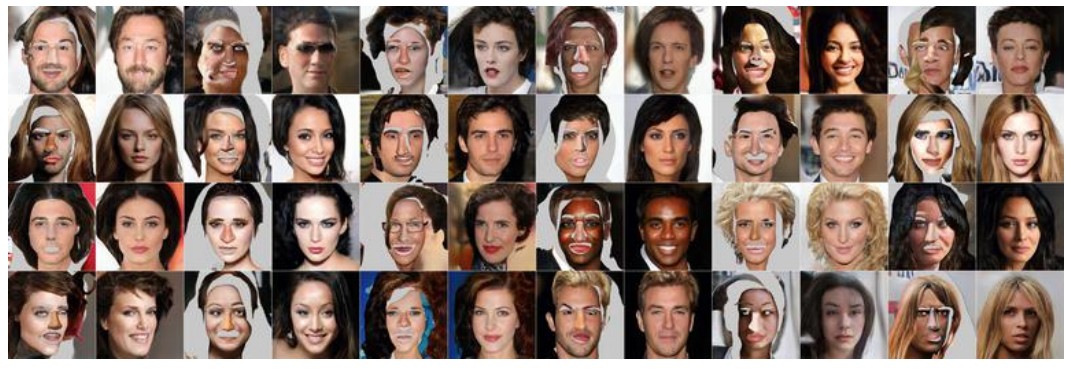

(a) ProGAN CelebA-HQ

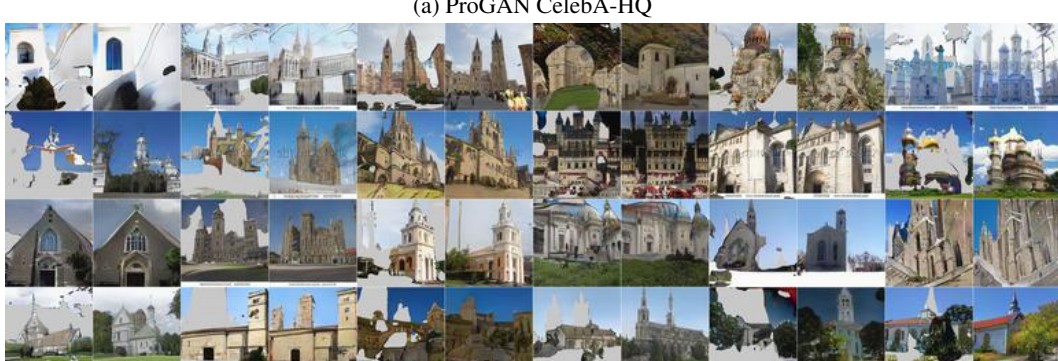

(b) ProGAN Church

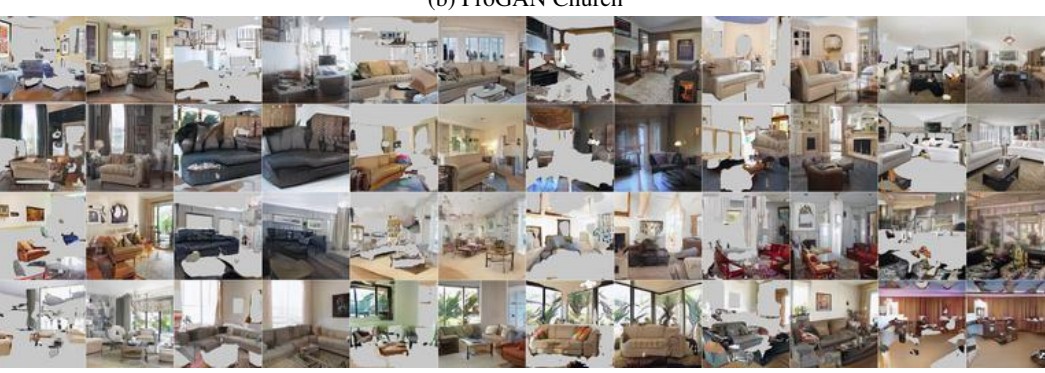

(c) ProGAN Living Room

Figure 24: Random samples of automatically generated colleges using a pretrained ProGAN generator.

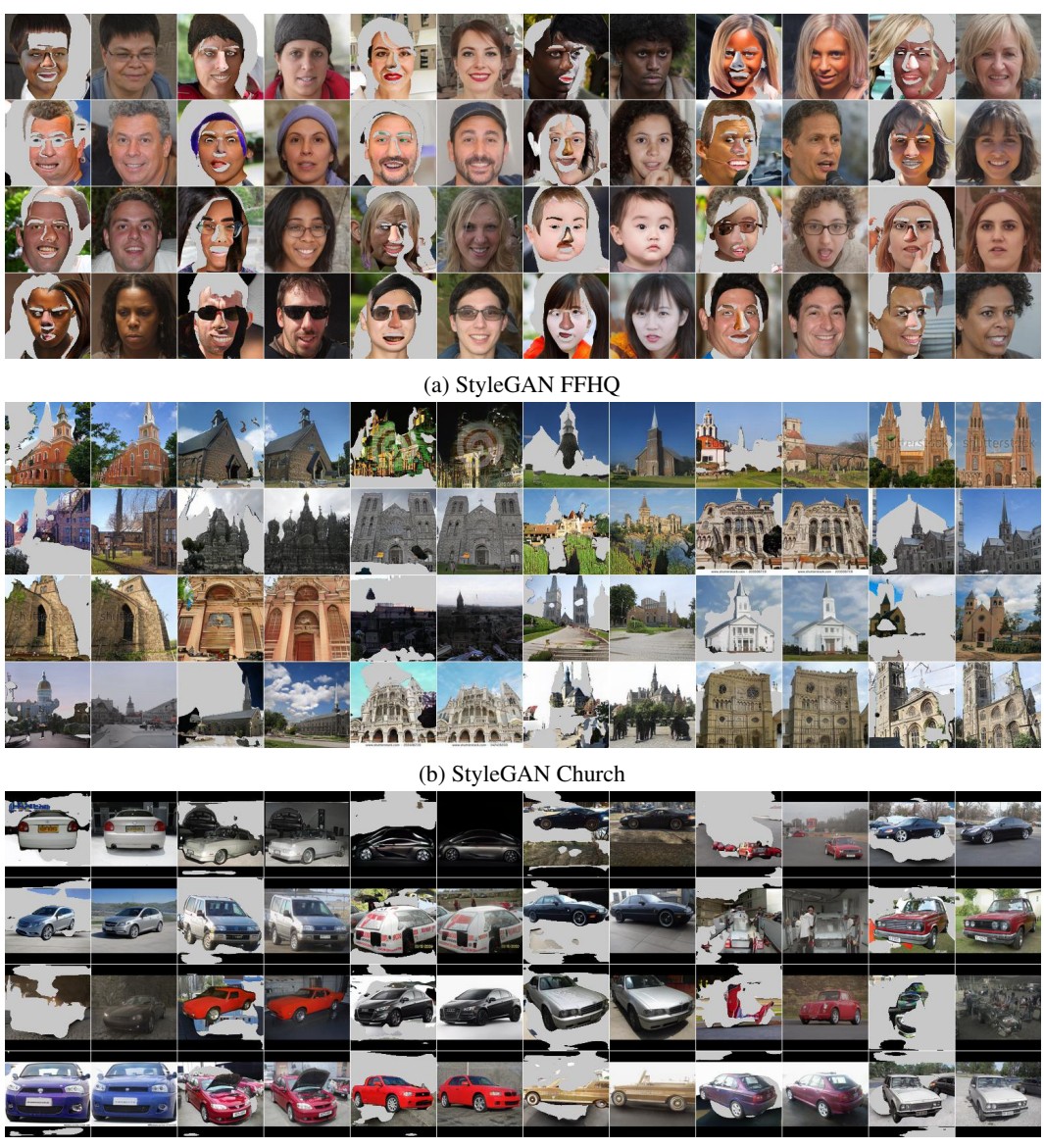

(a) StyleGAN FFHQ

(b) StyleGAN Church

(c) StyleGAN Car

Figure 25: Random samples of automatically generated colleges using a pretrained StyleGAN generator.

### A.2.5 ADDITIONAL PART VARIATION RESULTS

Here, we show additional qualitative results similar to those in Fig. 8 in the main paper. In each case, the heatmap shows appearance variation when changing the part marked in red. In Fig. 26, it can be seen that the variation is usually strongest in the face part that is changed, indicating that the composition of face parts learned by the model is a good match for our intuitive understanding of a face. In Fig. 27, we vary a single superpixel of a car. The resulting variations show regions of the images that commonly vary together (such as the floor, the body of the car, or the windows), which can be interpreted as a form of unsupervised part discovery.

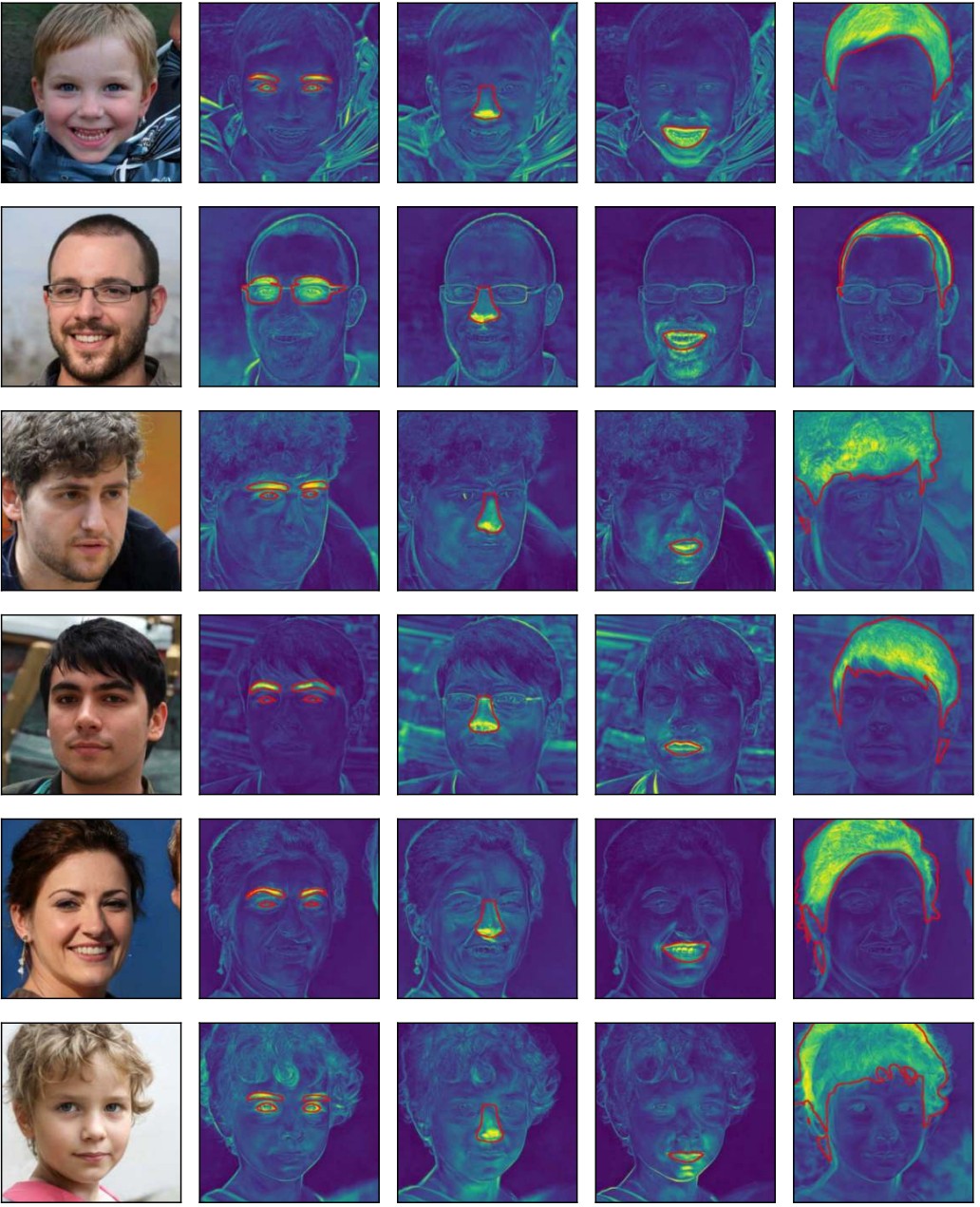

Figure 26: When changing semantic parts of a face (such as the eyes or the hair, shown in red), the resulting change in the image is often limited to those parts, indicating that the model parses faces into meaningful and intuitive components.

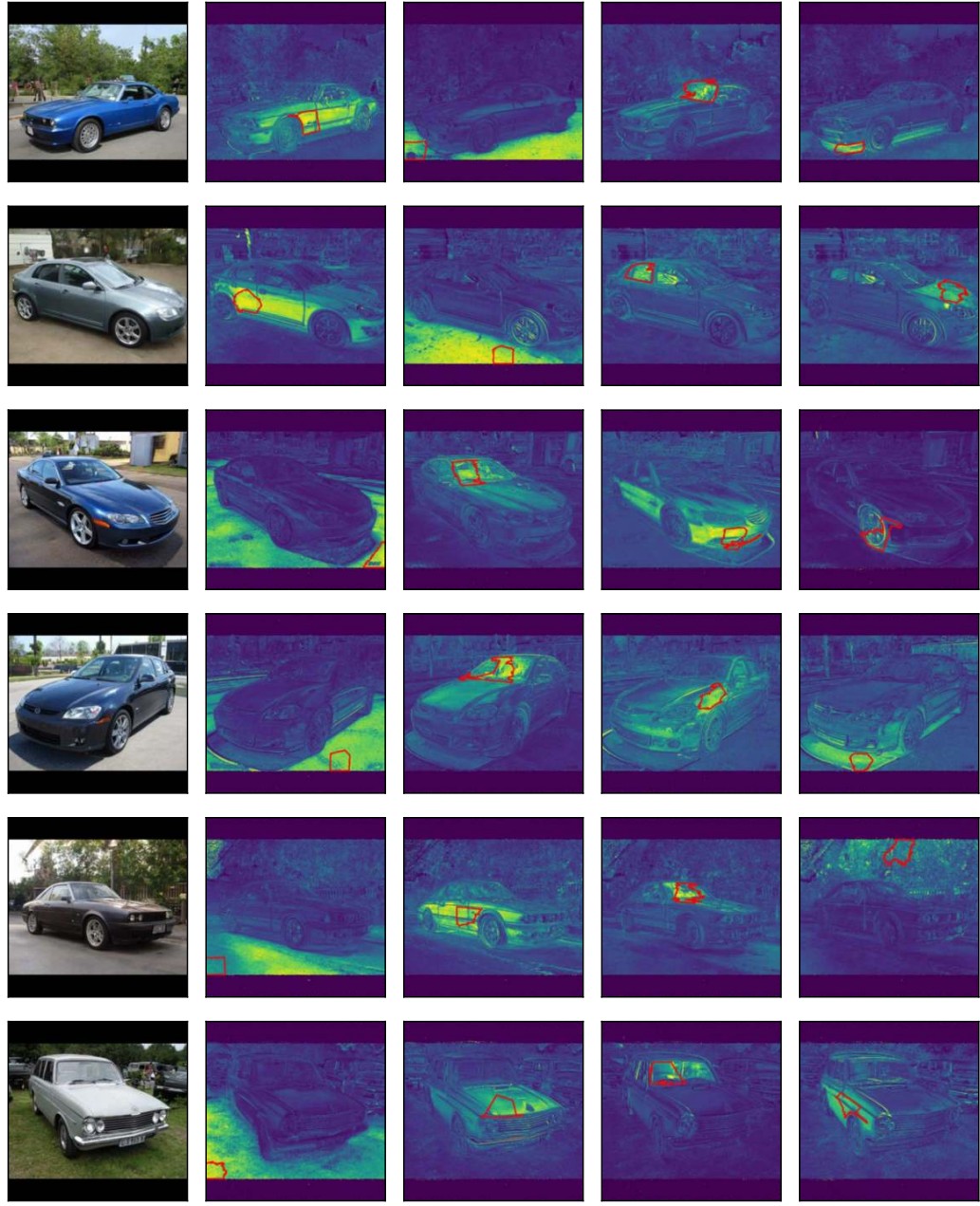

Figure 27: Our encoder-generator pair can be used to detect correlated parts of objects. Here, the value indicates how much a pixel changes when the superpixel shown in red is changed; oftentimes, semantic segments emerge, such as the car body, windows, the street, or the background.

