# OpenReview forum: "Using latent space regression to analyze and leverage compositionality in GANs"
_ICLR.cc/2021/Conference — ICLR 2021 Poster_

### Official Review · AnonReviewer3 · 2020-10-22
**Masked encoder seems like a great idea**

**Rating:** 7
**Confidence:** 4

**Review:**

I like this paper and I think it represents a very through-provoking and promising idea. The key aspect of this work is a (screen-space) masked encoder that learns to complete the loop with a previously trained generator. This allows for image completion, editing, collages, and essentially creates a prior for the generator’s training distribution. This allows the application to “snap” the (possibly partial) input to the manifold. While there are many encoders for GANs, I have never seen them formulated quite this way. The idea is also thoroughly compared and ablated.

Okay, with that out of the way, I can specify that these comments relate to the 22-page version including the supplement. The 8-page version is too terse and I found myself longing for details and discussions that were nowhere to be found (except in the supplement). For example, Eq. 4 is the beef, but it’s not really discussed in any meaningful way. Yeah, ok so you added a mask? But what does that _mean_? Given that now the encoder has only a random subset of pixels at its disposal and yet it needs to produce the full latent code, it would seem that it needs to learn very flexible ways of coming up with a plausible latent. Thus I think it starts to participate in generative modeling in a much more real sense than a typical full-image encoder does. As a reader I would like to see this kind of discussions on top of the mechanical descriptions.

Now, the real issue of course is the 8 page limit combined with a verbose layout, which suits many papers very well, but is a poor fit to multi-application works like this one. ICLR (and other conferences) should consider going to 10 pages with some “length must be proportional to the contribution” clause, because papers like this would be faster to write AND read if they were a bit longer. They would also be genuinely better. As an official reviewer I will read the supplement, but most others will not. Getting back to this paper: Obviously the authors have chosen the subset of material they felt was most compelling, and I won’t argue with that, but personally I didn’t like e.g. Sec 4.3 nearly as much as the discussion in A.2.1 (minus dataset rebalancing). I felt the former was unsurprising while the latter more clearly illustrated the key contribution, but that’s just a personal opinion. Sec 4.4 / Fig. 7 were very interesting for me, so please let those stay :)

As a summary, I find the idea clearly described, interesting, and likely to have many applications in future. Thus I recommend acceptance.

EDIT: Lowered score by one point after reviewer discussions. The evaluation could certainly be better and the difference to earlier mask-based methods more clear.

Minor:
- Fig 16: Are you sure your Poisson solver isn’t broken? I would have expected much better results in that column.
- Sec 4.4: Most of the time “c” is called component, but at least once it’s a “segment” instead. Should be component.
- Sec 4.3: “to obtained the blended image” —> “to obtain the blended image”

---

> ### Author Response · Authors · 2020-11-21
> **Response to Reviewer 3**
>
> We thank the reviewer for the helpful comments. We agree that parts of A.2.1 are a strong contribution, and we plan to use the additional page of text, which we are afforded in the final version, to move Fig. 8 and associated text to the main paper; we leave it in supplementary material for now to avoid numbering confusion with current reviewer comments and the updated pdf.
>
> **Masked regression objective**: We have added additional discussion in the methods section, and we thank the reviewer for their insightful comments here: “Intuitively, this masking operation on the encoder operates similarly to “dropout” (Srivastava et al.; 2014) on pixels -- it encourages the encoder to learn a flexible way of recovering a latent code that still enables the generator to reconstruct the image. Thus, given only partial images as input, the encoder is encouraged to map from these known pixels into a part of the latent space that is semantically consistent with the rest of the image (as deeper layers of the generator focus on semantic concepts). This allows the generator to reproduce an image that is both likely under its prior and consistent with the observed pixels."
>
> **Poisson blending**: While we used OpenCV seamless cloning originally for the Poisson blending, we have checked that against another independent implementation, which yields the same results. However, we note that the Poisson blending incrementally blends 4-8 images (depending on the domain) using the respective mask for each image -- the images are randomly sampled and there is no alignment between these layers, so the Poisson blending algorithm seems to struggle to compose these together realistically.
>
> **Minor comments**: We thank the reviewer for catching the typos; we have fixed these in the revised pdf.

---

### Official Review · AnonReviewer4 · 2020-10-23
**Thin evidence despite large number of experiments**

**Rating:** 5
**Confidence:** 4

**Review:**

The paper investigates controllable image synthesis using generative adversarial networks. The key component is a masked encoder that finds an approximate latent code given an image and a mask, so that the generated image matches the given image at the unmasked parts.

As a point of clarity, it is not always clear which experiments use generated images and which use real images. The use of Z loss when training the encoder of course requires the use of generated images, but using the encoder does not. The necessity of Z loss is left somewhat unclear even with the appendix, as is the need for the complex three-component loss in general. In the projection method presented in StyleGAN2 paper, only LPIPS loss is used and it appears to finds latent codes for generated images quite well.

I'm worried about regressing to W+ latent space with the StyleGAN variants. As is evident from, e.g., the bottom two rows in Figure 4 of Abdal et al., this latent space is so large that any generator can produce pretty much any image from this kind of latent code. In this view, the regressor network can be seen as just a conveniently weak bias towards natural-looking images - with a hypothetical perfect latent regression network, it should be possible to regress both generated and real images into W+ with fairly small resulting image error. This would presumably make the composites also retain all their artifacts instead of falling on the manifold of generated images. The imperfect nature of the regression network seems to be necessary for the image manipulation to succeed, but this is not addressed or analyzed in the paper.

The increase in FID shown in Figure 3 is quite remarkable. Compared to StyleGAN, the FFHQ and LSUN Car FIDs are over 10x worse than in the original. The reason for this would need to explored in more detail. Table 3 in A.2.4 reports some StyleGAN FIDs that do not match previously published numbers. FID of LSUN Car is 3.27 in the StyleGAN paper (2.32 in StyleGAN2), but Table 3 reports it as 22.06, almost 10x worse than state of the art. Reported FFHQ FID of 3.28 is between the StyleGAN papers but does not match either of them. The ProGAN FIDs are also consistently worse than reported in StyleGAN paper for the same datasets. These inconsistencies are puzzling, especially given that the paper claims to use pre-trained networks.

I'm also somewhat concerned that the greater realism in terms of density (Figure 5) may be explained simply by the regressor network's incapability to create latent codes that are not at the densest regions, and the much higher FIDs are in line with this. If so, the method requires a tricky balancing act between image quality and retaining of individual components in the composite. In one direction the image quality falls, and in the other direction the amount of control falls. It is unclear if there is a sweet spot where both are good, and the examples in the paper do not convince me that there is.

Overall the retaining of context does not seem to be very good. In Figure 10, the church image changes nearly completely except for the location of horizon, even though only the tower is supposed to change. The amount of control seems to be lacking as well. There is only one example where different modifications are done on the same base image (Figure 10 in Appendix A.2.1) and even there the laughing face acquires a mustache even though there is none in the composite. Experiments in Figure 12 and 13 also show that the composite is very different from the context image, and both are badly corrupted in all cases.

To summarize, when neither image quality nor control are properly demonstrated, I cannot see this method providing any advantages compared to previous work on editable GAN synthesis.

Pros: important topic, paper is well written.

Cons: quality of results is dubious, analysis of regressor network remains thin despite the large number of experiments, retaining of context with different modifications is not demonstrated

---

> ### Author Response · Authors · 2020-11-21
> **Response to Reviewer 4**
>
> We thank the reviewer for the helpful feedback. Below, we address the main points of the reviewer.
>
> **Imperfect regression**: It is true that imperfect regression is necessary for the image-prior effect to occur. In fact, when we randomly combine image parts as a collage, we do not desire to perfectly reproduce the input - this would actually be harmful as the inputs are not realistic. Instead, we sacrifice some reconstruction accuracy, trading it off for global consistency in the reconstructed output while maintaining the spirit of the input. In Sec 4.1 (Fig. 3) we compare this tradeoff across different generator domains, while in Sec 4.2 (Fig. 5), we compare for different model setups; we find that minimizing L1 in fact leads to lower realism on random input collages. Additionally, we seek to analyze GANs as a scientific object, where in fact failure to retain context indicates some limitations in compositionality (Fig. 7; comparison of ProGAN and StyleGAN). While the encoder maintains high density, we also report coverage metrics in Table 3, which with the exception of FFHQ faces are similar between the encoder reconstructions and the GAN samples. This indicates that the encoder does not just cover high-density regions but covers regions of the data distribution to a comparable degree as the standard generator. The reconstruction of a specific input can further be improved by optimizing the encoder’s initial latent code; however, we find that a forward pass of the encoder is sufficient to allow interesting image completion and image composition effects to emerge from a pretrained generator.
>
> **FID**: We thank the reviewer for pointing out this discrepancy. For the ProGAN generators, we have used a pytorch version of these models (from the repository of [1]) for training the encoder, which are different from the tensorflow generators in [2]. These pytorch models have slightly higher FID (we can replicate the FID reported in [2] using samples from the tensorflow generator and the same FID computation script). Secondly, the FID is quite sensitive to intermediate resizing of the image, thus we have removed any resizing prior to the FID computation. Third, for the cars domain, the GAN generates images with border padding; we added identical padding to real images, following [3]. This enables us better reproduce the reported numbers on the GAN sample FID, we have updated numbers accordingly in Table 3.
>
> **StyleGAN2 Projection**: The key difference between the StyleGAN projection method and ours is inference time -- using an encoder is nearly instantaneous (allowing for large scale analysis experiments and graphics applications), while the projection method takes over a minute of optimization per image. However, we add the StyleGAN projection method in our revised pdf (Fig 5 and 17). Qualitatively, the StyleGAN projection method yields artifacts in the missing regions of the input, and the optimization loses semantic priors that allow the encoder/generator combination to generate a globally consistent image. Quantitatively, the StyleGAN projection can achieve lower L1 than the pure encoder methods, but subsequently optimizing the output of the encoder yields similar L1 to StyleGAN projection, yet higher density.
>
> |             | StyleGAN Projection | ProGAN Encoder | StyleGAN Encoder | ProGAN Encoder + Optimize | StyleGAN Encoder + Optimize |
> |-------------|----|----|----|----|----|
> | L1    | 0.228  | 0.317  | 0.300  | 0.229  | 0.145  |
> | Density | 0.250  | 0.785  | 2.077  | 0.884  | 0.377  |
>
> **Real or generated images?**: All experiments in the main text are done on generated images, except for Sec 4.2 which compares the same images across different generator architectures -- in this case, rather than using images from one generator, we extract parts of real images so that no generator has an unfair starting advantage. Our aim in this paper is to investigate the capabilities and priors of a pretrained GAN in recombining its concepts -- for example, if it can create the same building but with different sky scenery -- rather than focusing specifically on real image editing. As such, our encoder is trained only on generated images; while we have also experimented with training the encoder on real images in conjunction with generated images, we found that this did not have a big impact on the encoder performance for our purposes. There are small differences in image reconstruction when using different loss combinations, but the encoders have the same basic behavior when used in conjunction with the generator: they map incomplete and inconsistent inputs to a reasonable part of the latent code, so that the GAN consequently generates a realistic image.
>
> [1] Bau et al. “Seeing what a GAN cannot generate.” ICCV 2019
>
> [2] Karras et al. “Progressive Growing of GANs for Improved Quality, Stability, and Variation.” ICLR 2018
>
> [3] Karras et al. “Analyzing and Improving the Quality of Stylegan.” CVPR 2020

---

### Official Review · AnonReviewer1 · 2020-10-29
**Impressive Image Editing Results**

**Rating:** 8
**Confidence:** 4

**Review:**

**PAPER SUMMARY**

The paper trains feedforward networks to project input images into the latent space of a pretrained GAN generator, and shows how this can be used both for various image editing tasks as well as to probe the internals of the trained generator. Experiments demonstrate that this can be used for tasks such as image composition, image completion, attribute modification, and multimodal image editing.

**STRENGTHS**

- The paper is well-written and easy to follow
- The proposed method is both simple and effective for a variety of tasks
- Experimental results (both qualitative and quantitative) are impressive and thorough
- Extensive supplementary material providing additional details to aid reproduction

**WEAKNESSES**

- Some missing references to BiGANs / ALI

**MISSING REFERENCES**

One missing line of related work is that of BiGANs [1, 2] / Adversarially Learned Inference [3] in which an inference network is jointly trained with the generator and discriminator to project samples into the latent space. However the overall goal of these papers is often some kind of unsupervised feature learning, which is very different from the image editing applications presented in this submission.

[1] Donahue et al, “Adversarial Feature Learning”, ICLR 2017

[2] Donahue and Simonyan, “Large Scale Adversarial Representation Learning”, NeurIPS 2019

[3] Dumoulin et al, “Adversarially Learned Inference”, ICLR 2017

**OVERALL**

On the whole this is a strong paper. The method is simple and effective, the results are impressive, the experiments are thorough, and the paper is very well-written and easy to follow. This is a clear accept in my view.

**AFTER REBUTTAL**

After reading the other review's and the author responses, my opinion is unchanged: This is a well-written paper with a simple and effective method, and a clear accept.

---

> ### Author Response · Authors · 2020-11-21
> **Response to Reviewer 1**
>
> We thank the reviewer for the helpful feedback.  Preliminary experiments on BigBIGAN[1] (where the encoder is trained jointly with the generator) showed that oftentimes the reconstruction does not preserve properties of the input -- e.g., if we blend a dog and a grassy field, the output yields a different dog in a grassy field. On the other hand, while the recent BigGAN generator achieves stunning image quality, current methods for finding the latent code corresponding to a given image tend to require many iterations of optimization [2], making these methods substantially slower than a feed-forward encoder.
>
> We thank the reviewer for the suggestions of the additional references, and we have updated our related works section accordingly: “Rather than inverting into pre-trained generator, one can train the generator and encoder jointly; based on modifications to the Variational Autoencoder (Kingma & Welling, 2014);  Donahue et al. (2016), Donahue & Simonyan (2019), Dumoulin et al. (2016) use this setup to investigate the properties of latent space representations learned during training, while Pidhorsky et al. (2020) demonstrates a joint learning method that can achieve comparable image quality to recent GAN models.”
>
> [1] Donahue and Simonyan, “Large Scale Adversarial Representation Learning”, NeurIPS 2019
>
> [2] Huh et al., “Transforming and Projecting Images into Class-conditional Generative Networks”, ECCV 2020.

---

### Official Review · AnonReviewer2 · 2020-10-29
**Lack of novelty, weak evaluation**

**Rating:** 5
**Confidence:** 4

**Review:**

In this paper, the authors propose a latent space regression method for analyzing and manipulating the latent space of pre-trained GAN models. Unlike existing optimization-based methods, an explicit latent code regressor is learned to map the input to the latent space. The authors apply this approach to several applications: image composition, attribute modification, image completion, and multimodal editing. They also present some analysis on the independence of semantic parts of an image.

This paper addresses an interesting problem of manipulating the latent space of pre-trained GANs. However, learning an explicit latent regressor (encoder) is already explored in [1]. According to [1], the biggest problem of learning such an encoder is the encoded latent code cannot fully reconstruct the original content. As can be seen in image completion in Fig. 1, the contents in the inverted image are perceptually different from the input. This is due to the limitation of expressiveness of the latent space, which is discussed in recent works including [1]. Based on it, technical novelty of this paper is significantly limited and I could not find the discussion about it in this paper.

The authors discuss about image inversion methods (GAN prior methods such as [1]), but did not compare them. State-of-the-art works need to be compared to verify the effectiveness of the proposed method. It looks like existing methods are also applicable to applications addressed in this paper.

[1] Gu et al., Image Processing Using Multi-Code GAN Prior. In CVPR, 2020

After rebuttal, the technical novelty is still not convincing. The masked encoder is widely used in image inpainting works. Combining multiple existing techniques may work better than existing methods, but it has little impact on the community. Thus, I do not change my rating.

---

> ### Author Response · Authors · 2020-11-21
> **Response to Reviewer 2**
>
> We thank the reviewer for the helpful comments.
>
> To the revised pdf we add additional comparisons to a number of previous works: (1) Multi-code inversion [1] which blends multiple latent codes at an intermediate layer, (2) StyleGAN2 Projection [2] which directly optimizes to the StyleGAN W space (we use the W+ option), and (3) and In-domain inversion [3] which uses an encoder to regularize subsequent iterative inversion. However, there are a few key differences between these works and ours.
>
> Namely, the methods of Multi-Code [1] and Stylegan2 projection [2] are based on iterative optimization. We modify both methods to use the appropriate masked loss. Iterative methods are much slower than a single feed forward pass of our encoder, which is a tradeoff: optimization can better match the input image but is much slower. On the other hand, we find that iterative optimization can destroy semantic priors when minimizing reconstruction error, such as the “floating tower” in Fig 17. On the same set of 200 random church collages we use in Sec. 4.2 and A.2.4, we add these two methods as additional comparisons. Quantitatively, like the other optimization-based methods we compare to, [2] and [3] can achieve low L1, but density is also low, signalling that the inversion of these images is not as realistic. Qualitatively, we find that [2] leads to blurry results on the composite, while [3] results in inpainting artifacts. We summarize these statistics below, and have updated Fig 17 and Fig 5.
>
> |             | ProGAN Encoder | StyleGAN Encoder | Multi-Code Prior | StyleGAN Projection |
> |-------------|----|----|----|----|
> | L1  | 0.317  | 0.300 | 0.128  | 0.228  |
> | Density | 0.785  | 2.077  | 0.206  | 0.250  |
> | Inference Time    | 0.021  | 0.072  | 477.932  | 86.873  |
>
> The In-domain [3] method is similar to ours by using an encoder, but unlike ours, the encoder does not use a masked objective. This means that the encoder cannot distinguish between pixels to reconstruct and pixels to fill in; whereas we find that adding this inpainting objective helps exploit the GAN’s priors when given unrealistic inputs. [3] also provides a diffusion approach, with further optimizes only within a masked region based on the initial encoder prediction. As [3] does not provide a pretrained encoder on churches, we try both the encoder and the diffusion method on our set of face collages (in Fig 5. and Fig 18). Qualitatively, using the in-domain encoder creates artifacts due to the missing pixels in the random composites, and the secondary diffusion step further overfits to the misalignments in the inputs. Quantitatively, both the in-domain encoder and diffusion method have lower density (are less realistic), although the diffusion method can attain lower masked L1 distance due to the subsequent optimization. We summarize the results in the table below, and have updated Fig 5. and Fig 18.
>
> |             | ProGAN Encoder | StyleGAN Encoder | In-Domain Encoder | In-Domain Diffusion |
> |-------------|----|----|----|----|
> | L1  | 0.265  | 0.198 | 0.208  | 0.124  |
> | Density | 1.66  | 2.01  | 0.443  | 0.403  |
>
> We use our encoder method to investigate emergent priors of the latent code once a partial image or an unrealistic collage of random image parts is projected into latent space. There is an inherent trade-off here, as in the case of the random collages, we do not desire to perfectly reproduce the input, but change it just enough so that the output becomes a realistic image while still maintaining the spirit of the input. For example, if we collage parts of two images with different sky color, it may be desirable to harmonize them to the same color (higher L1 reconstruction), or we maintain the inconsistency to attain lower L1, but less realism. Our results show that this encoder-decoder serves as a regularizer to pull these images closer to the manifold of realistic images. Moreover, our method only relies on a forward pass and is fast enough for interactive applications, but of course similarity to the input image can be improved with further steps of per-image optimization.
>
> [1] Gu et al. “Image processing using multi-code gan prior.” CVPR 2020
>
> [2] Karras et al. “Analyzing and improving the image quality of stylegan.” CVPR 2020
>
> [3] Zhu et al. "In-domain gan inversion for real image editing." ECCV2020

---

### Decision · Program_Chairs · 2021-01-07
**Final Decision**

**Decision:**

Accept (Poster)

**Comment:**

The scores here are bimodal.
The low-scoring reviewers have problems with the evaluation, and I agree it could be improved.
The high scoring reviewers seem to mostly agree with those complaints, but think that the paper is interesting enough
to be accepted anyway.
One of the low-scoring reviewers has some complaints about novelty that I don't find super convincing.
The other low-scoring reviewer has suggested that they'd be OK with a decision of Accept.

Part of me thinks that I should reject this paper with a message of "come back later with the experiments improved", and that
that would be the best thing for the field, because the paper can already be publicized on arXiv anyway.
But the other part of me thinks: what if they do that and get unlucky with a bad batch of reviews the next time (the current reviewers were great and had a really thorough discussion)?
With some amount of trepidation, I'm recommending accept, but *please* reward my faith in you (the authors) and make an effort to fix the things reviewers complained about before the camera ready.

---

> ### Author Response · Authors · 2021-01-27
> **Proposed evaluations before camera ready**
>
> Thanks to the AC and anonymous reviewers for the helpful discussion. In our revised pdf, we have added additional comparisons to multi-code and in-domain inversion as suggested by R1, and fixed the FID discrepancy as pointed out by R4. Before camera ready, we additionally plan to address:
> - improving the reconstruction quality of the StyleGAN encoder
> - add comparisons and discussion on image inpainting methods
> - investigate FID, in addition to the density metric evaluated in submission, on the composition testbed comparing different methods
>
> Please let us know if there are any other suggested evaluations, we are happy to investigate.